# Satellite Imagery for Rapid Detection of Liquefaction Surface Manifestations: The Case Study of Türkiye–Syria 2023 Earthquakes

**Maria Taftsoglou** [1] , **Sotiris Valkaniotis** [1,*] , **George Papathanassiou** [2] **and Efstratios Karantanellis** [3]

1 Department of Civil Engineering, Democritus University of Thrace, University Campus, 671 00 Xanthi, Greece; mtaftsog@civil.duth.gr
2 Department of Geology, Aristotle University of Thessaloniki, 541 24 Thessaloniki, Greece; gpapatha@geo.auth.gr
3 Department of Earth and Environmental Sciences, University of Michigan, 1100 North University Avenue, Ann Arbor, MI 48109-1005, USA; stratis@umich.edu
* Correspondence: svalkani@civil.duth.gr

**Abstract:** The 6 February 2023 earthquake doublet (Mw 7.7 and Mw 7.6) that occurred on the East Anatolian Fault Zone (EAFZ) triggered a significant amount of soil liquefaction phenomena in SE Türkiye and NW Syria. The great areal extent of the affected area and the necessity of rapid response led to the adoption and improvement of a workflow for mapping liquefaction phenomena based on remote sensing data. Using satellite imagery, we identified 1850 sites with liquefaction manifestation and lateral spreading deformation. We acquired a thorough map of earthquake-triggered liquefaction based on visual mapping with optical satellite imagery (high and very high-resolution) and the aid of radar satellite imagery and interferometry. The majority of sites are found along meandering sections of river valleys, coastal plains, drained lakes, swamps, and lacustrine basins along the East Anatolian Fault, highlighting once again the influence of geomorphology/surficial geology on the distribution of liquefaction phenomena. A total of 95% of the liquefaction occurrences were mapped within 25 km from the surface trace of the fault, confirming the distance from fault rupture as a more effective tool for predicting the distribution of liquefaction than epicentral distance. Thus, taking into consideration the rapid documentation of these phenomena without the limitations in terms of time, cost, and accessibility of the field investigation techniques, this desktop-based approach can result in a rapid and comprehensive map of liquefaction from a strong earthquake, and can also be used as a future guide for subsequent field investigations for liquefaction hazard mapping.

**Keywords:** liquefaction; lateral spreading; Turkey–Syria earthquakes; satellite imagery; interferometry; geomorphology

## 1. Introduction

Soil liquefaction occurs when a strong ground shaking changes the arrangement of saturated silty and fine sandy soil particles and increases the pore water pressure [1–3]. The consequences of liquefaction include the ejection of a mixture of water and sand at the ground surface resulting in a ground surface subsidence, and lateral spreading phenomena. The recent earthquake-induced liquefaction phenomena, triggered by the 2010–2011 Canterbury earthquake sequence in New Zealand [4,5] and the 2012 Emilia earthquake sequences in Italy [6,7] indicated that liquefaction can be the major cause of severe economic losses affecting the resilience of the community.

Documentation of liquefaction case histories is one of the most important steps for understanding the mechanism of liquefaction triggering and has significantly contributed to the mitigation of the liquefaction-induced failure models [8,9]. Rapid estimates of the effects of an earthquake can help improve emergency response and minimize induced

casualties [10]. Post-earthquake liquefaction data collection traditionally relies on field investigation. However, in cases of large seismic events, field-based mapping is less efficient in terms of time and cost, where the liquefaction area is too extended and can be sporadically due to the inaccessibility of some locations [11]. Furthermore, the short lifespan of surface liquefaction manifestations holds back their in situ detection and requires immediate response from the reconnaissance team. In order to establish a site investigation plan and improve the efficiency and impact of field reconnaissance trips, remote sensing techniques are applied to preliminary document the liquefaction sites [12].

Remote sensing techniques range from simple visual/manual classification methods to semi-automated thematic classification and change detection techniques [13]. Liquefaction can be identified at a regional scale through optical and Synthetic Aperture Radar (SAR) satellite imagery or at a site-specific level using aerial imagery from UAVs. Optical data analysis provides a two-dimensional image of the affected area after the earthquake, which can be used to document the spatial distribution of phenomena across a region. On the other hand, SAR data can be acquired regardless of the sun illumination and cloud coverage and allow analytical techniques such as radar interferometry (InSAR). Characteristic examples of using remote sensing tools to detect liquefaction-induced surface disruption are the Haiti 2010 [14], Tohoku, Japan 2011 [15], Christchurch, New Zealand 2010–2011 [16], Oklahoma, US 2011 [17] and Damasi, Greece 2021 [12] events where ejecta, craters, fissures, and lateral spreading phenomena were mapped by visually looking at the satellite and aerial imageries.

Moreover, an approach that is proposed for the detection of liquefaction manifestations is one that is based on the automatic comparison of the pre- and post-event imagery, measuring the differences based on specific indicators of soil moisture (multi-temporal change detection process). After the Bhuj 2001 earthquake in India, Ramakrishnan et al. [18] used this method to detect the liquefaction phenomena by combining different spectral bands from Indian Remote Sensing Satellite (IRS-1C) and using the indicator of wetness index for soil moisture. In addition, Oommen et al. [11] related the surficial manifestations of liquefaction phenomena with the changes in temperature and water content of the Earth's surface, using pre- and post-Landsat imagery. Furthermore, the detection of liquefaction phenomena using Synthetic Aperture Radar (SAR) images is mainly based on changes in coherence values between pre- and post-earthquake images by Papathanassiou et al. [12]. Without being affected by atmospheric conditions, SAR is highly sensitive to various surface changes. As was shown by recent case studies, liquefied sites/areas identified by SAR techniques are in good agreement with the ones delineated by traditional field-based methods [15,16,19,20].

The goal of this study is twofold: to detect the liquefaction manifestations that were widespread along the rupture zone of the February events and to preliminary assess the correlation of geomorphological features with location and clusters of liquefaction phenomena. The latter issue was initially investigated by Youd and Perkins [21] who proposed the relevant criteria of liquefaction susceptibility, while many researchers recently highlighted this strong correlation concluding that liquefaction manifestations are not randomly distributed at the epicentral area, but are mainly concentrated at specific depositional environments [22–29]. In order to achieve this, an extensive remote survey took place, preliminarily focusing on areas close to the earthquake rupture, with low relief values, covered by Holocene and Quaternary formations of fluvial, lacustrine, and coastal sediments. As a result, a map of liquefaction surface manifestations, i.e., ejecta and lateral spreading sites are presented and discussed in the relevant sections at the end of this paper.

### 1.1. The 6 February 2023 Türkiye/Syria Earthquake Doublet

A large number of destructive earthquakes struck EAFZ in the last 2000 years due to the dynamic motion of relative plates with slip rates of $10 \pm 1$ mm/yr [30]. According to seismic catalogs [31–35], at least four historical earthquakes were related to secondary effects such as ground openings, sand ejections, and river slumping, causing severe distraction both to the urban and free field areas. The first three earthquakes occurred near Antioch, with the oldest

event dated 14 September 458 AD (M$_s$ 6.5) and the other two on 29 March 526 AD (M$_s$ 6.8) and 29 November 1114 AD (M$_s$ 6.8). On 13 August 1822, a M$_s$ 7.4 earthquake occurred in Southeastern Anatolia and damaged Gaziantep and Antakya in Turkey and Aleppo and Han Sheikhun in Northwestern Syria. Most significant instrumental earthquakes along the EAFZ include the 1964 Malatya Ms 5.7, the 1971 Bingol Ms 6.9, the 1986 Dogansehir, Malatya Ms 5.9, the 2003 Bingol Mw 6.3, and the 2004 Sivrice Mw 5.5 one. The 2010 Kovancilar, Elazig earthquake of Mw 6.1 was triggered by the activation of a ~30 km fault at the northeastern extent of the EAFZ [36], while the last largest event before the 6 February 2023 was the 2020 Mw 6.8 Sivrice earthquake that ruptured around 45 km of the EAFZ [37–39]. Other major earthquakes include the 1975 Lice Ms 6.7 and 1992 Erzincan Mw 6.7 events which were accompanied by many significant aftershocks on the highly stressed segments of nearby North Anatolian Fault Zone and Southeast Anatolian Thrust Zone.

The first event Mw 7.7 occurred on 6 February 2023 at 04:17 (01:17 GMT) (AFAD) at Türkiye's Pazarcık district in Kahramanmaraş with a focal depth of 8.6 km. The coordinates of the earthquake's epicenter are N37.288°, E37.043° (AFAD). The initial event occurred on a splay fault structure below Narli Basin near Pazarcik, and then the rupture expanded across the main EAFZ [40,41]. The rupture speed of this event was calculated to be 3.2–3.3 km/s [40,42] with surface displacements on the order of 3–7 m. The spatial distribution of aftershocks (Figure 1) indicates that the earthquake rupture reached Antakya (Hatay) in the south and come to an end in the north at the Pütürge segment, close to the Doğanyol, Elazığ earthquake rupture of 2020 [38,39]. The total rupture length was measured as 310–350 km [43–45] with a surface rupture mapped length of 270 km [46].

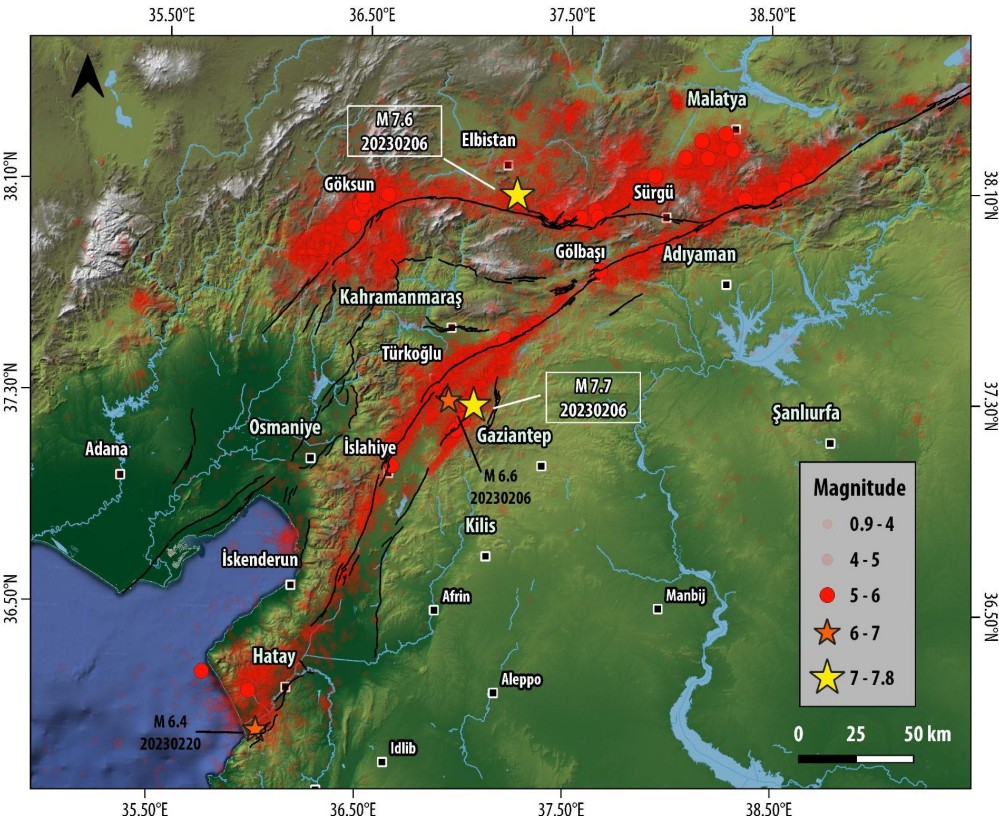

**Figure 1.** Aftershocks for the M7.7 and M7.6 earthquakes. Epicenter locations derived from AFAD catalog (6 February 2023–2 May 2023).

A second major event (Elbistan earthquake) took place a few hours later at 13:24 (10:24 GMT) in Ekinözü, Kahramanmaraş with Mw 7.6 and a focal depth of 7.0 km (AFAD). The epicenter of the second earthquake was located at N38.089°, E37.239° (AFAD), close to

the mapped trace of the Sürgü Fault (SF). The rupture speed for this event was calculated at 2.5–2.8 km/s [40,42]. The length of the second rupture was 150–170 km with the surface displacements on the order of 2–8 m [44,45]. Analyses of teleseismic data for both cases revealed almost pure left-lateral strike-slip motion on nearly vertical fault, while the two earthquakes were considered as a "twin", due to their comparable size and the different fault structures they were detected [40].

The earthquakes affected the provinces of Kahramanmaraş, Adıyaman, Hatay, Osmaniye, Gaziantep, Kilis, Şanlıurfa, Diyarbakır, Malatya, Adana, and Elazığ, causing more than 50,000 human casualties in Turkey and Syria. Several segments of transportation and energy infrastructures were severely damaged [46,47]. The number of collapsed buildings or the ones that must be demolished reached 156,000 [48].

Extensive earthquake-induced phenomena were documented such as surface ruptures (primary effects) and liquefaction, rockfalls, and landslides (secondary effects). Among these effects, surface ruptures and soil liquefaction were the predominant phenomena that caused failures both in free fields and in manmade environments. An impressive length of primary surface fault ruptures was documented shortly after the earthquakes, along the ruptured segments of EAFZ [49–52]. Major rockfalls and landslides were detected in the area of Adıyaman and Islahiye (Fevzipaşa train station in Gaziantep), where rolling boulders caused damage to the railroad and road network. Furthermore, extensive liquefaction was observed at the shores of lakes in Gölbaşı (Adıyaman), in the İskenderun Port wharf area, and in Antakya near the Asi River [49].

### 1.2. Geological Setting

The epicenters of the two strong earthquakes were recorded in the East Anatolian Fault Zone (EAFZ), one of the major active tectonic structures of the Eastern Mediterranean region. The northeast–southwest trending EAFZ constitutes the 580 km long southeastern boundary of the westward-moving Anatolian Block [53,54] and is associated with frequent shallow seismicity in the top ~20–25 km of the crust and left-lateral strike-slip type tectonism [38,39,55,56].

As the East Anatolian Fault Zone continues to the southwest, it is divided into several distinct geometric segments based on fault step-overs, jogs, or changes in fault strike comprising different pull-apart basins and uplift zones rather than one continuous surface through fracture [54,57–60]. The main southern EAF strand extends from Karliova to Antakya and links the Dead Sea Fault Zone (DSFZ) and the Cyprus Arc (CA) around the Amik triple junction. It consists of seven fault segments, namely from NE to SW as Karliova, Ilica, Palu, Pütürge, Erkenek, Pazarcik and Amanos Fault segments. The 350 km long northern strand is forming between Çelikhan and the Gulf of Iskenderun, connecting the Sürgü–Misis Fault system (SMF) with Kyrenia–Misis Fault Zone (KMF) [54,60,61].

The ruptured southern part of the main strand of the EAFZ is formed by Erkenek, Pazarcik and Amanos Fault segments. Following a left-lateral strike-slip movement, the fault dissects mountainous terrain and valley slopes westwards, with Late Pleistocene and Holocene sediments concentrated close to Çelikhan and in Gölbaşı pull-apart valley. Pazarcik segment extends between Gölbaşı and Türkoğlu releasing stepovers covered by Holocene and Quaternary deposits. The Amanos Fault segment begins at Lake Gavur as Nurdağı section and ends in İslahiye depression filled with Quaternary alluvial deposits and basaltic lavas. Continuing from the Nurdağı segment, the Hassa Fault joins the area between İslahiye releasing bend and Demrek restraining stepover, while the last section of Kirikhan Fault terminates near Topboğazı. Through these three fault sections of 115 km in length, the Amanos segment transects basement rocks, Quaternary basalts, and alluvial deposits [54].

In the northern strand of the EAFZ, west of Çelikhan, the Sürgü segment traverses through a valley created by a 17 km long shutter ridge, covered by alluvial deposits. Then the fault continues for 20 km westward along the southern branch of the Sürgü River Valley, where tributaries and intervening ridges are systematically offset and finally merge in the

Nurhak area. The Çardak segment on the other side joins the area between the Nurhak and the Göksun and bifurcates into two sections, the eastern and the western separated by a 500 m wide stepover [54].

Karasu Valley forms an NNE-trending topographic depression rising from <100 m in the south near Antakya to almost 500 m altitude in the north near Kahramanmaras [62]. It is bounded to the west by the southernmost tip of EAF, Amanos Fault, and to the east by the northernmost segment of DSF, Yesemek Fault. Both marginal faults are left-lateral strike-slip and overlap each other with a left stepover along the trough. The Karasu River flows through basaltic and alluvial formations: the first section concludes the NW Saglik and NE Narli Plains, which are drained by the Aksu River and are separated by a hilly E–W trending landscape. The middle part of the Karasu trough is a narrow hilly corridor of ophiolitic rocks, bordered by Quaternary lavas. The last and southern section includes the Amik Basin and Orontes (Asi) River Valley, close to the Syrian border.

The majority of Holocene and Quaternary sediments are found in pull-apart and extensional basins along the main strands of EAFZ (Figure 2). These consist mainly of the Holocene riverbed, floodplain, marsh, lacustrine and coastal plain deposits, Holocene and Quaternary fans, and Quaternary river terraces and volcanics. Under favorable conditions, such as strong ground motion and elevated groundwater surfaces, many of these unconsolidated sediments are susceptible to liquefaction.

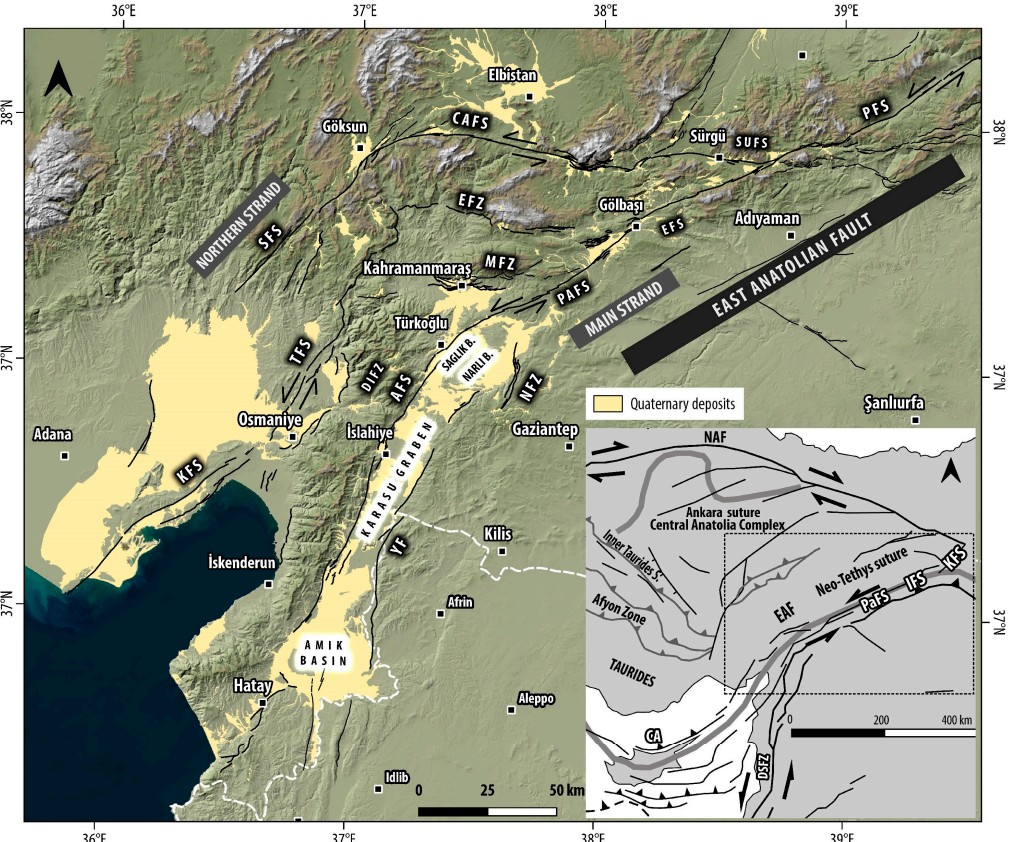

**Figure 2.** Active fault map of Eastern Anatolia showing the East Anatolia Fault segments (black lines, from [54]). Holocene and Quaternary deposits from [63]. Inset box at lower right shows the regional tectonic setting modified after [64]. KFS: Karliova Fault segment, IFS: Ilica Fault segment, PaFS: Palu Fault segment, DSFZ: Dead Sea Fault Zone, CA: Cyprus Arc. PFS: Pütürge Fault segment, EFS: Erkenek Fault segment, PAFS: Pazarcik Fault segments, AFS: Amanos Fault segments, NFZ: Narli Fault zone, YF: Yesemek Fault, SUFS: Sürgü Fault segment, CAFS: Çardak Fault segment, SFS: Savrun Fault segment, EFZ: Enginek Fault zone, TFS: Toprakkale Fault segment, KFS: Karataş Fault segment, MFZ: Maraş Fault zone, DIFZ: Düziçi–İskenderun Fault zone.

## 2. Materials and Methods

### *2.1. Data*

As has been previously mentioned, one of the goals of this study is to delineate within a short period after the earthquake's occurrence, the manifestations of possible liquefaction based on desktop (remote survey) studies. In order to achieve this, a methodology proposed by [12] was applied. In particular, the studies that must be performed during a post-earthquake survey are separated into phase I and phase II. The first phase is related to remote survey studies, realized a few hours or a few days after the occurrence of the mainshock, aiming to (i) preliminary assess the likely liquefaction areas based on the empirical relationships correlating the earthquake magnitude to the distance of liquefaction surface manifestations, (ii) collect basic information considering historical liquefaction occurrences and geology/geomorphology and (iii) reveal the extent of liquefaction areas based on pre- and post-event satellite imageries and associated interferograms. Having detected the liquefaction zones, the second phase (phase II) can be realized, consisting of a field-based reconnaissance survey including ground-truthing techniques.

2.1.1. Optical Satellite Imagery

For identifying and mapping liquefaction manifestations from the 6 February 2023 earthquakes, we used mostly optical satellite imagery that was acquired in the first days and up to four weeks after the event (Figure 3).

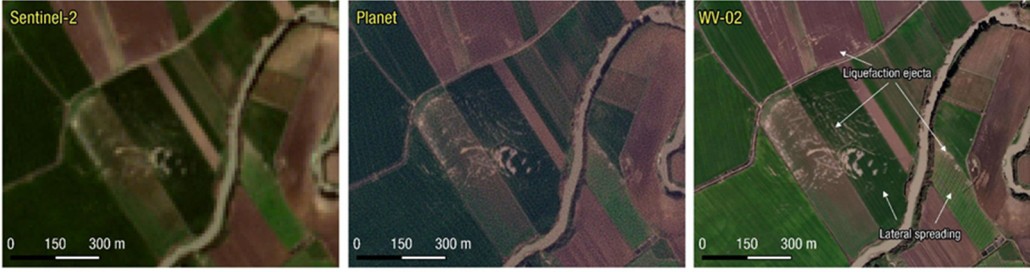

**Figure 3.** Comparison of liquefaction features in different optical sensors and resolutions. From left to right; (**Sentinel-2**) (10 m), (**Planetscope**) (3–4 m), (**Maxar WorldView-02**) (0.46 m).

Copernicus Sentinel-2 multispectral imagery with an acquisition swath of 290 km and 10 m spatial resolution (for red, blue, green, and near-infrared bands) offered a suitable overview of the affected area. While the resolution is not optimal, the coverage of individual acquisitions and frequent revisits (3–5 days) can provide a rapid dataset to identify major manifestations of liquefaction-related phenomena. We used Sentinel-2 images as the first step in identifying sites of interest. Flooding of the plains and the upper course of Orontes River Valley in the south, near Antakya (Figure S1), and snow coverage of the northern areas prohibited the use of semi-automated methods for liquefaction mapping, and only visual mapping was performed. Water extruded from craters or fissures and saturated liquefaction ejecta can be identified by utilizing near-infrared and short-wave infrared bands of Sentinel-2 as surface moisture and water. But the presence of flood water and snow makes the distinction between flood-related and liquefaction-related water hard to distinguish; thus, we avoided the use of multi-spectral band ratios and pseudo-color composites to identify liquefaction-related phenomena. Sentinel-2 imagery was successfully exploited for rapid remote mapping of liquefaction phenomena in recent earthquakes such as the 2020 Mw 5.8 Lone Pine, CA earthquake [65] and 2021 Mw 6.3 Thessaly, Greece earthquake [12,66].

The next step after the exploitation of Sentinel-2 imagery was acquisition of Planet optical imagery for a significant part of the affected area (excluding areas with snow cover to the north). Image acquisitions of the Planetscope constellation (with an acquisition swath of 24–32 km and 3–4 m spatial resolution) for 2 February, 8 February, and 13 February were obtained and examined. Coverage map of Planet imagery is presented in the supplementary section (Figure S2). A number of individual frames (from both Planetscope and SkySat

constellation sensors) in various spots were also published by Planet as open data and obtained through the OpenAerialMap portal.

The majority of our mapping was performed with very high-resolution (VHR) optical satellite imagery that covered a significant part of the earthquake rupture and affected areas. A large number of VHR frames acquired shortly after the earthquakes (from 7 February to 5 March) were released by Maxar through their Open Data Program (Disaster Response Geospatial Analytics). Images were acquired by GeoEye-1 and WorldView 1-2-3 sensors and were published in the Analysis-Ready Data (ARD) format. ARD format involves pre-processing of the raw imagery such as atmospheric compensation, orthorectification, pansharpening, and radiometric balancing. Maxar VHR images have a spatial resolution of 0.3–0.7 m, depending on the sensor. A coverage map of Maxar VHR imagery is presented in the supplementary section (Figure S3).

### 2.1.2. SAR Satellite Imagery

In addition to optical imagery, we also exploited Synthetic Aperture Radar (SAR) satellite imagery for identification and mapping of liquefaction manifestations. Copernicus Sentinel-1A images with an acquisition swath of 250 km and 5 m by 20 m spatial resolution were available covering the whole study area. Multiple frames of both ascending and descending orbits were needed to cover the affected area, with an acquisition interval of 12 days for each frame and orbit (as of December 2021 due to the malfunction and decommission of Sentinel-1B satellite). Despite the fact that radar imagery does not offer the same imaging capabilities as optical imagery, it can be exploited through InSAR (Interferometric Synthetic Aperture Radar) processing in order to identify major land cover changes, and also provide detailed maps of ground surface deformation. Frame id, dates, and coverage map are included in the supplementary section (Figure S4 and Table S1).

### 2.1.3. Other Sources

A number of other imagery sources were also available during the post-earthquake period for detecting liquefaction phenomena. Aerial and satellite mapping of the area around the earthquake ruptures was available from the Turkish Ministry of National Defense, General Directorate of Mapping. These orthophotos were used to fill in areas not covered by Maxar and Planet VHR imagery. Additionally, several UAS (unmanned aircraft systems) orthophoto surveys of settlements affected by the earthquakes from various sources, were published through the OpenAerialMap portal.

For the examination of local geomorphological features that might be masked by modern human activity (croplands, urban expansion), we used declassified KH-4 Corona optical satellite imagery. Those images, dated in 1960s–1970s, provide a good overview of surficial fluvial features that might not be visible today due to intense irrigation farming and land reclamation [67,68]. Orthorectified frames of selected dates were obtained from the CORONA Atlas of the Middle East project [69]. Frame id and dates of Corona frames used are included in the Supplementary Materials (Table S2).

### 2.2. Methodology

#### 2.2.1. Selection of Inspection Area

The 6 February 2023 earthquakes originated from two long fault ruptures of ~350 km for the Mw 7.8 Pazarcik event and ~160 km for the Mw 7.6 Elbistan event. Overall area affected was estimated to be larger than 80,000 km$^2$; thus, a workflow focusing on the identification and mapping of earthquake-induced liquefaction phenomena was necessary (Figure 4).

Based on previous experience in rapid mapping of liquefaction phenomena [12], we targeted our search using the published ShakeMap by USGS as a guide, keeping the area roughly enclosed by the 0.1 g peak ground acceleration (PGA) contour. Uncertainties and generalization of ShakeMap products drive the selection of a conservative value for PGA fencing, to account for local effects and simplified fault source modeling. Our research

started from the area along the surface fault trace, which was preliminary mapped using optical imagery, co-seismic displacement maps, and InSAR. The locations of earthquake-induced features such as surface fault rupture and fracture openings are considered sites prone to detect ejection of liquefied material and zones where the ground motion is expected to be very strong, particularly in proximity to the fault rupture.

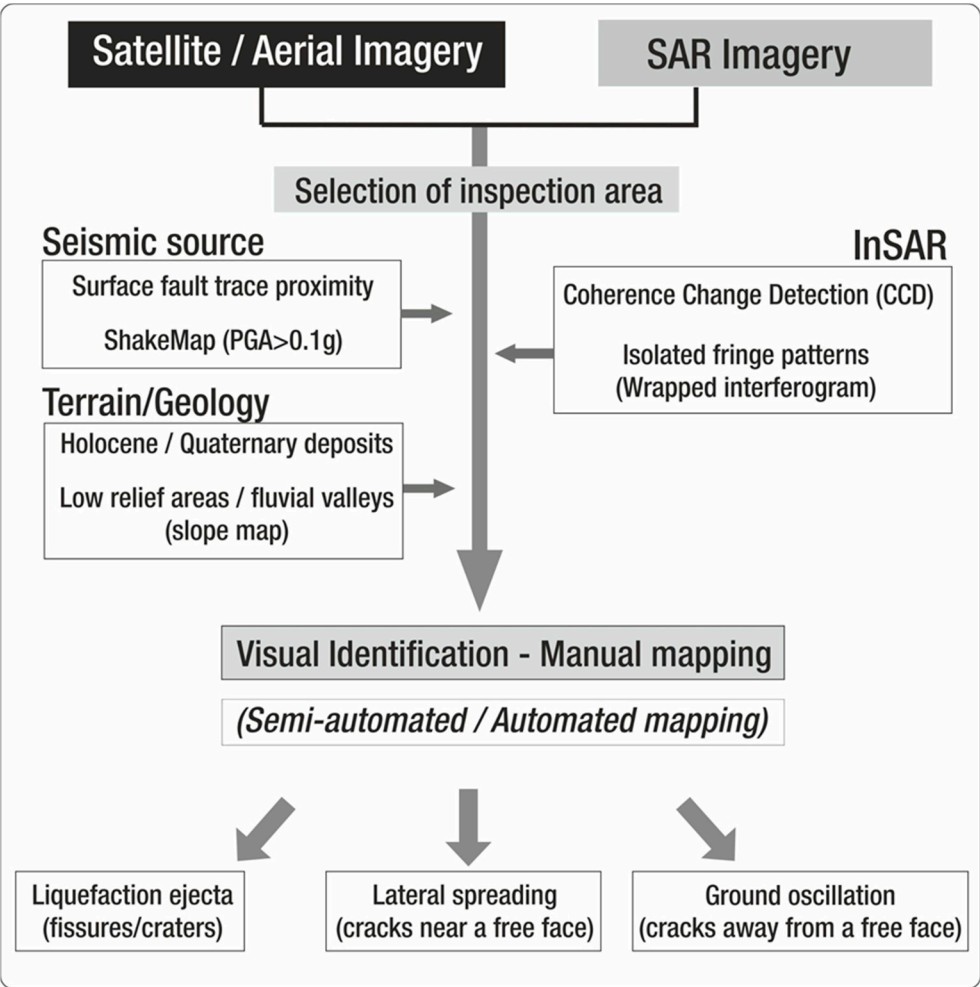

**Figure 4.** Analytical flow chart of steps for identification and mapping of liquefaction-related phenomena using remote sensing data.

The next step of our research was the identification of susceptible to liquefaction areas close to the earthquake rupture. Using available regional geological maps, we were able to delineate Holocene and Quaternary surficial geological formations and exclude older ones, i.e., Pre-Quaternary age, and bedrock areas. Recent non-cohesive fine to medium grain sediments like coastal deposits, fluvial deposits, alluvial plains, and drained lakes/swamps are susceptible to liquefaction. These are also the formations that are most likely to be saturated with a shallow groundwater table. Since Holocene and Quaternary mapped formations might also include coarse-grained or cohesive sediments and geological formations like debris/scree, we further screened these areas using digital elevation models. Low relief areas, meandering river valleys, enclosed basins, pull-apart basins along the Eastern Anatolian Fault, and narrow fluvial valleys with terraces were primarily selected for research. Areas with slopes larger than 3 degrees were excluded as they might involve more cohesive and less susceptible sediments, such as alluvial fans and cemented deposits. Further visual screening excluded, for example, deep entrenched riverbed valleys and arid plains with hard arid rock pavement.

2.2.2. InSAR Analysis

While high-resolution optical images like Sentinel-2 and Planet can offer a quick assessment of major surface manifestations of earthquake-induced liquefaction phenomena, interferometry (InSAR) can also map and pinpoint locations where liquefaction and lateral spreading displacement occurred.

We processed Sentinel-1 interferometric pairs using Alaska Satellite Facility's (ASF) Hybrid Pluggable Processing Pipeline (HyP3) for production of wrapped (phase) interferograms and line-of-sight (LOS) displacement maps, and European Space Agency's Sentinel Application Platform (SNAP) software for extracting coherence maps.

Significant liquefaction and lateral spreading deformation can be identified through InSAR as (a) isolated and complex fringe patterns away from fault ruptures in the wrapped interferograms and (b) concentrations of coherence loss through coherence change detection (CCD). Complex fringe patterns not connected to primary or secondary fault deformation are attributed and related to ground oscillation or large-scale lateral spreading displacement [70]. These types of fringe patterns were identified in co-seismic wrapped interferograms in Hatay Province area (mouth and alluvial valley of Orontes River and Amik Plain) and also in Narli Basin (Figure 5).

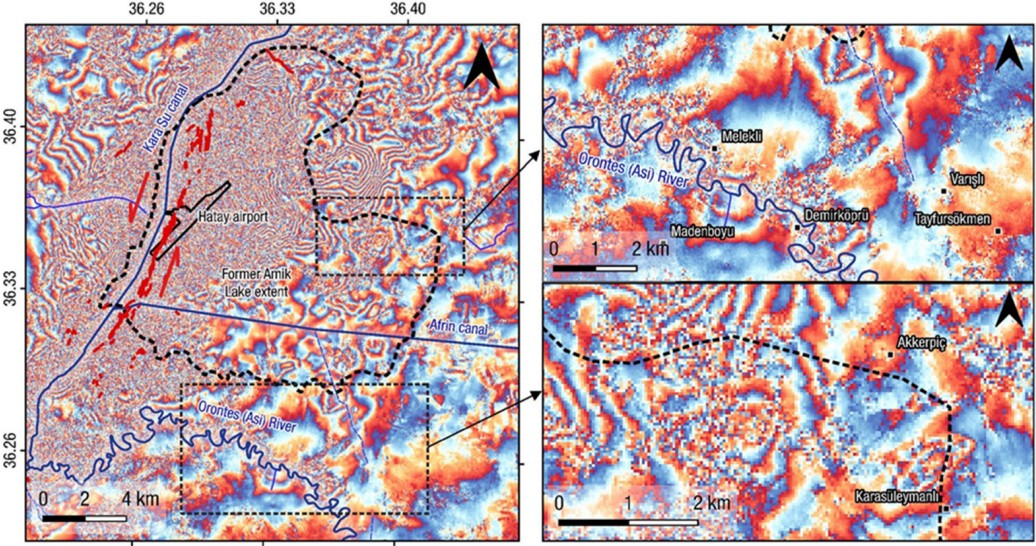

**Figure 5.** Co-seismic phase interferogram (Sentinel-1 ascending track, 28 January 2023–9 February 2023) over Amik Plain, Hatay Province; irregular and isolated fringe patterns in the eastern part of the basin away from the main fault deformation (western part) correspond to spots with significant liquefaction deformation such as subsidence, lateral spreading, or ground oscillation across the former Amik Lake floor (dotted line).

Interferometric coherence is a proxy for severe surface disturbance/land cover change and/or strong localized displacement. Large areas with coherence loss might involve large-scale lateral spreading displacement and multiple surface manifestations of earthquake-induced liquefaction [12,15–17]. Concentrations of coherence loss also occur along the surface fault trace of an earthquake rupture, and it might conceal local major liquefaction spots. Moving further away from the fault rupture, major areas of coherence loss can be attributed to liquefaction and lateral spreading, after comparison with pre-event interferometric pair coherence maps to exclude the possibility of temporal land cover changes or other non-earthquake sources of disturbance. An example of the results produced from coherence change detection for the 6 February earthquakes in Amik Plain, Hatay Province, is shown in Figure 6.

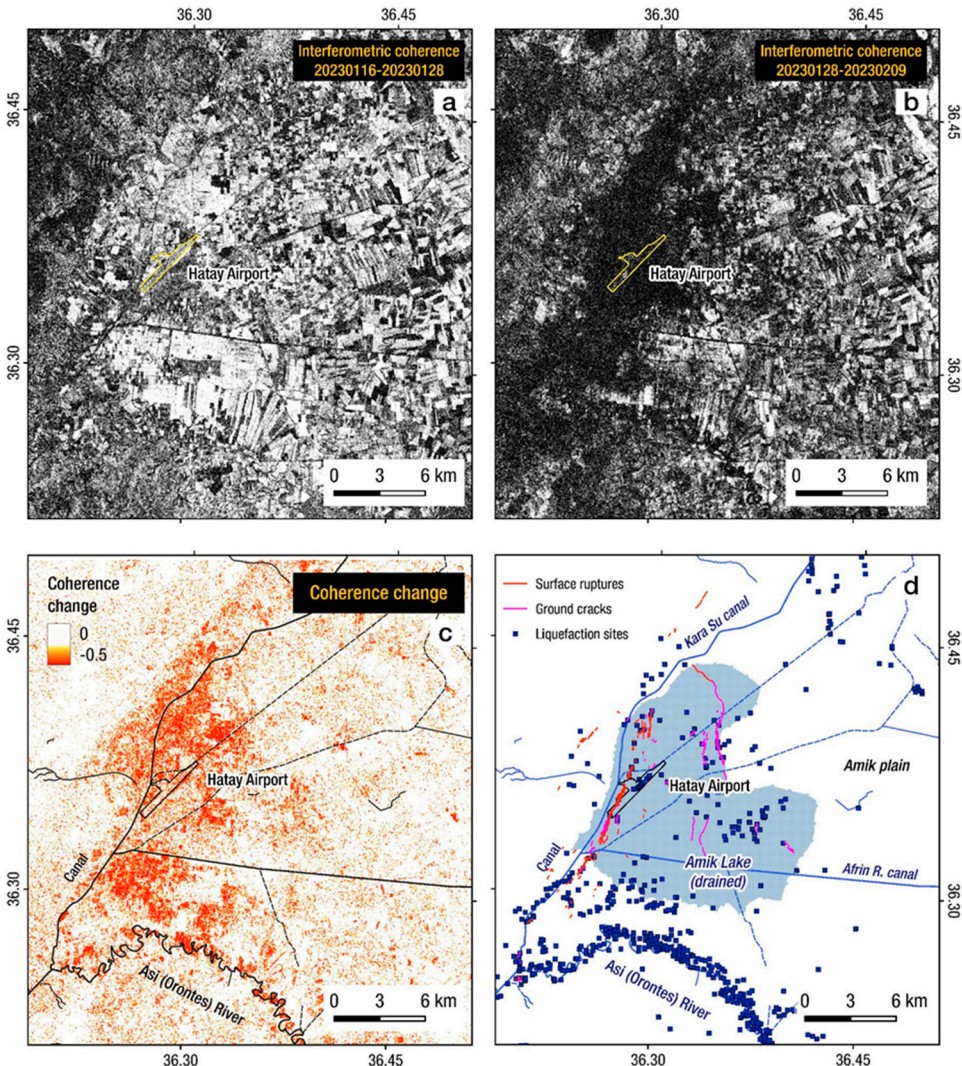

**Figure 6.** Interferometric coherence over Amik Plain, Hatay Province; pre-earthquake pair (**a**) in comparison with co-seismic pair (**b**). Dark areas mark significant coherence loss. Coherence change is shown in (**c**). Concentrations of coherence loss coincide with areas with significant liquefaction manifestations (**d**) like sand blows, lateral spreading, and ground oscillation across the former Amik Lake floor.

### 2.2.3. Visual Mapping

Mapping of liquefaction manifestations was performed manually, using the inspection and focus methodology described previously and graphically shown in the flow chart of Figure 4. The same workflow can be applied for either visual identification and manual mapping or by semi-automated techniques and processing of imagery. Regarding the 6 February 2023 earthquakes, considering the (a) limitations imposed by significant snow cover at the northern sections of the study area, (b) steep terrain in large sections, and (c) presence of irrigated crop areas with significant seasonal and temporal land cover change, we excluded the use of semi-automated methods of mapping and adopted a manual approach.

The imagery was examined following the Sentinel-2-->Planet-->VHR sequence, with gradually increasing detail and smaller feature identification. Site locations were mapped as points representing either isolated features or concentrations (clustering) of multiple liquefaction features. Points along areas of multiple features were located roughly every few tens or hundreds of meters depending on the distribution of features. Considering the regional scale dimensions of this study, we adopted this method for rapid reconnaissance

and mapping instead of focusing on detailed delineation of every singular feature in each site, which is more time-consuming and does not affect the regional distribution of the inventory. Mapping of identified individual liquefaction features is in progress and the relevant map will be released in near future. The compiled map of liquefaction site locations is provided in the supplementary section.

Visual identification of earthquake-induced liquefaction phenomena was based on the characteristic features that appear on aerial and satellite imagery [13,20,71] (see examples in Figure 7). Identified features were cross-checked using as many pre-event images as possible, aiming to exclude the presence of these features before the earthquake and disqualify possible non-earthquake-related land cover changes. Irrigated crops were especially inspected for the pre-event presence of patterns that can be a source of false positive identifications [72]. These patterns or spots in cropland that could not be validated in the pre-event timeline were excluded from mapping. Liquefaction manifestations on the surface included mostly linear fissures, circular sand boils/craters, and ejecta along fault ruptures or ground cracks, either isolated or in complex groups of multiple features (clusters). Deformation by lateral spreading and ground oscillation was manifested by parallel ground fractures and cracks, slides towards rivers and canal banks in the former case, and patterns of ground cracks away from free faces in the latter one. Most liquefaction phenomena were concentrated along meandering river valleys, abandoned strands or avulsions of rivers, drained lake basins and swamps, and coastal areas.

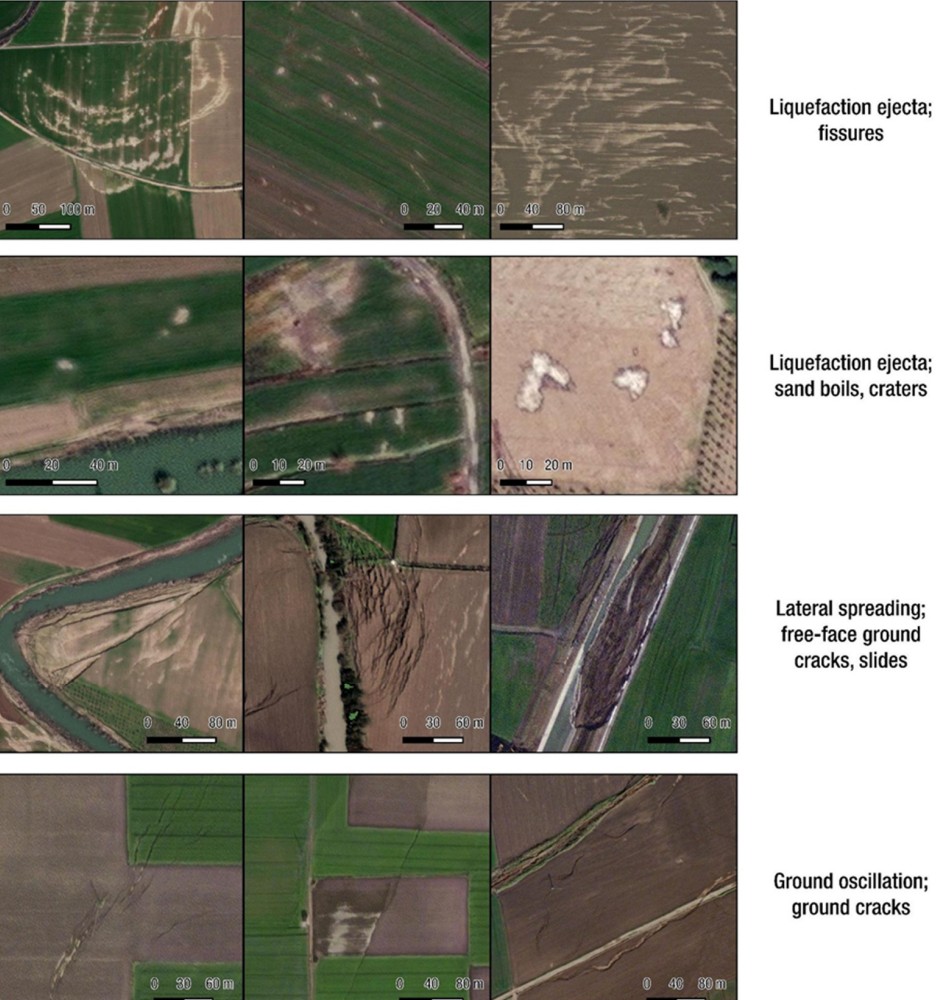

**Figure 7.** Examples of different types of earthquake-induced liquefaction phenomena identified and mapped using optical imagery.

## 3. Results

### 3.1. Spatial Distribution of Liquefaction Manifestations

Applying the aforementioned methodology, we identified 1850 sites of liquefaction and lateral spreading features along the EAFZ (Figure 8). Most of these manifestations were mapped in proximity to the surface rupture of the first M7.7 event. The second (northern) M7.6 earthquake rupture affected areas of higher latitude covered by snow, which limited the detection of liquefaction effects. The northernmost mapped sites are located north of Malatya, while the southernmost are in the mouth and valley of the Orontes River, in Hatay Province.

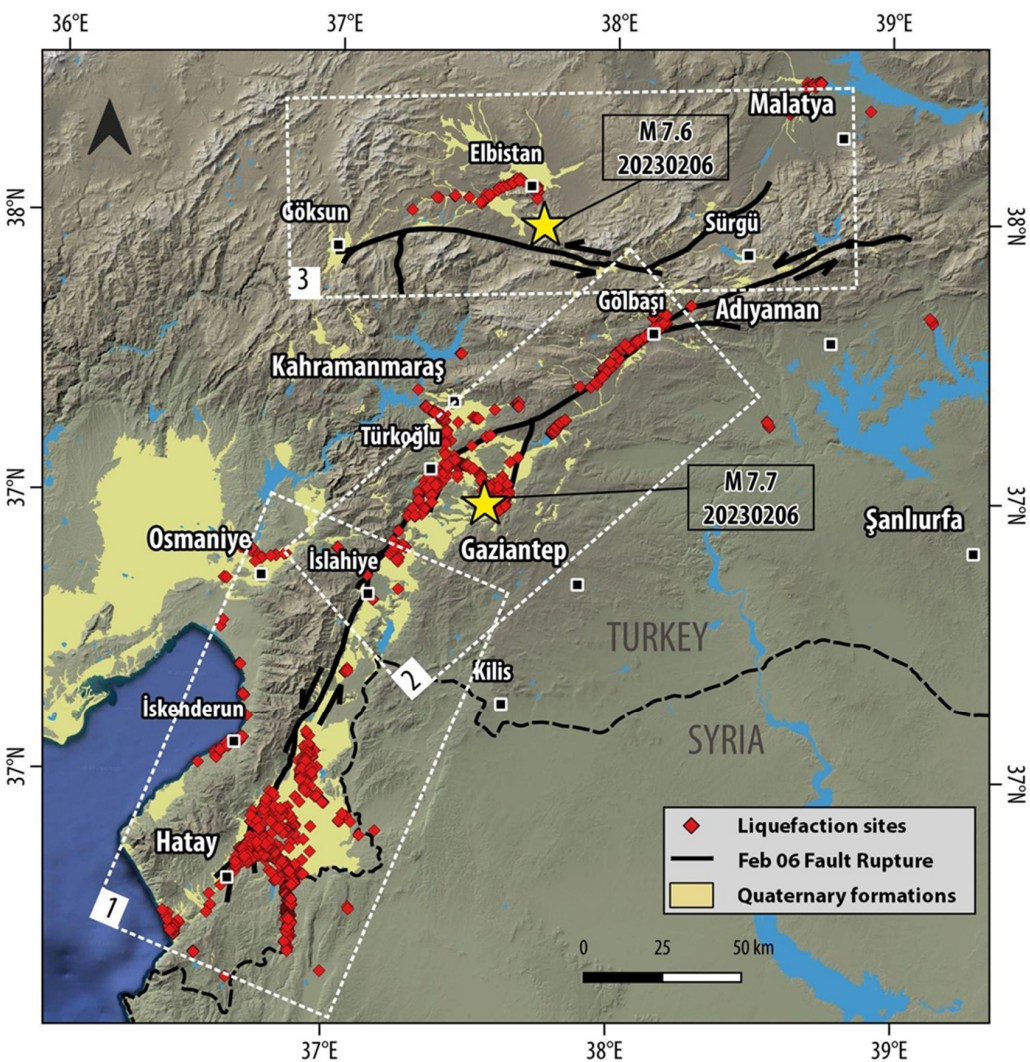

**Figure 8.** Overview map of liquefaction and lateral spreading sites identified and mapped using satellite imagery. Fault rupture is shown with black lines, as mapped from Sentinel-2 imagery and [52]. Epicenters of the February 6th seismic events are marked with yellow stars (AFAD). Quaternary formations along the fault rupture from [63]. Inset maps 1–3 are described in Figure 9.

In particular, craters and fissures with ejecta and lateral spreading phenomena were manifested in sites close to the fault rupture covered by fluvial, lacustrine, and coastal Holocene deposits (Figure 8). High-density areas of liquefaction phenomena are mapped from the southwestern coastal zone of Antakya and Iskenderun port up to the Northeastern Gölbasi Basin along the M7.7 fault rupture of the main strand of the East Anatolian Fault Zone. Sporadic manifestations were mapped in the northern region of Elbistan and Malatya and were considered to be triggered by the second M7.6 earthquake.

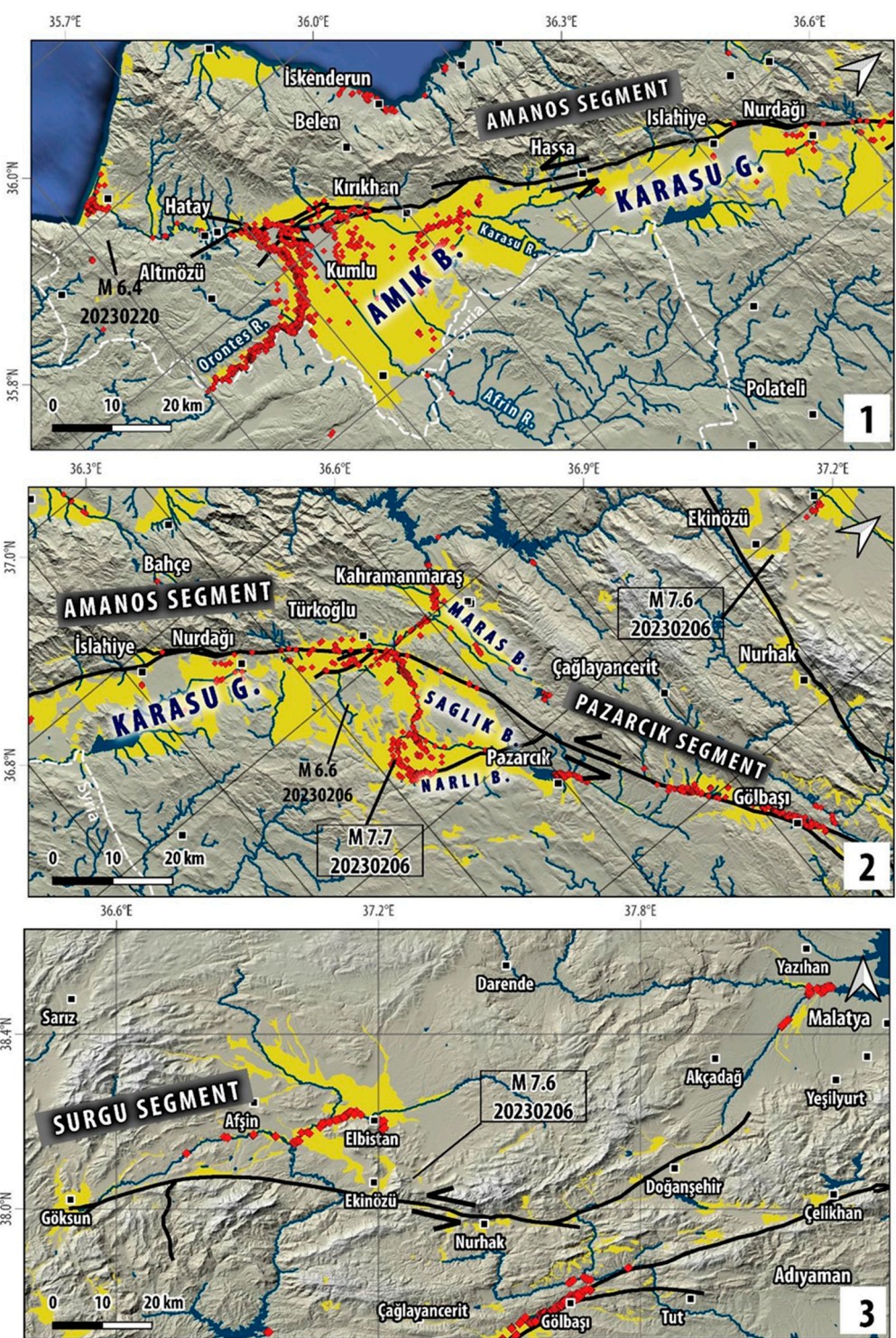

**Figure 9.** Detailed inset maps (**1–3**) along the East Anatolian Fault Zone, where liquefaction and lateral spreading phenomena were identified for the 6 February 2023 earthquakes. Fault rupture is shown with black lines, as mapped from Sentinel-2 imagery [52]. Epicenters of the 6 February seismic events are marked with yellow stars and orange stars for the 20 February M6.4 aftershock near Antakya (AFAD). Quaternary formations (yellow) along the fault rupture from [63]. Location of maps (**1–3**) is shown in previous Figure 8.

A significant number of liquefaction phenomena were identified south of Kahramanmaras City along the Karasu trough, close to the epicenter of the M7.7 earthquake. The northern section of Karasu Valley consists of Narli and Saglik Basins (Figure 9), with Maraş

Basin to the northwest. Most liquefaction and lateral spreading sites follow the Aksu River meander sections and the abandoned meanders through these basins. Numerous liquefaction phenomena were detected and mapped on the Pazarcik Dam lake floor and upstream towards the east. Approximately 20% of the total liquefaction records were mapped through the area (Figure 9).

The highest density of liquefaction manifestations was mapped north of Antakya, at Amik (Amuq) Valley and Orontes (Asi) River (Figure 10). Amik Basin is developed at the triple junction of the Amanos segment (EAF), Hacipasa segment (DSF), and Antakya segment (CA) [73–75]. Being infilled with Plio-Quaternary sediments of more than 200–300 m thickness, Amik Plain extends for approximately 36 km long and 40 km wide [76] and is drained by Orontes, Karasu, and Afrin Rivers from the south, north, and east, respectively. The western central part of the plain was formerly covered by Amik Lake. During the last decades, Amik Lake was completely drained [77,78] through an artificial channel system (the Balıkgölü Canal) into the Orontes River. After the first earthquake on 6th February, about 35% of the total liquefaction sites were detected within Amik Basin; a significant number of those concentrated along the central western and southern parts of the valley. Extensive co-seismic surface ruptures were also mapped through the westernmost part of the Hatay airport and the main water canal at the western section of the basin that collects Karasu River water through the basin. Using VHR images, we observed a vast accumulation of flood water through the area, covering more than 5 km$^2$. The eastern part of the airport was severely damaged by the liquefaction of the earth-fill sections of the runways, leading to multiple failures and its closure for 6 days.

The Orontes (Asi) River course constitutes nearly 50 km of the border between Turkey and Syria and exits to the south through Antakya Valley. During the 6 February seismic events, a significant number of craters and fissures with ejecta material and lateral spreading phenomena occurred and were detected by our study in the meandering, delta zones, and open valley sections of the Orontes River, despite the fact that a section of the river valley along the Turkey–Syria border was masked by flooding during the first days after the earthquakes (Figure S1). Very few sites were identified by remote sensing inside the Antakya urban region and the entrenched sections of the Orontes (Asi) River along Antakya Valley. Entrenched river valley sections limit the spatial extent of saturated non-cohesive surficial sediments, while free field liquefaction manifestations inside the Antakya urban area were limited or impossible to visually identify due to dense urban environment and the large volume of debris from building damages. However, despite the lack of results from remote sensing, indications of extensive sub-surface liquefaction in Antakya City were described by post-earthquake field surveys [49]. We managed to map 563 liquefaction sites along Orontes River Valley, which account for almost 30% of the total liquefaction records.

The city of Iskenderun was one of the most affected urban centers, due to liquefaction phenomena after the 2023 seismic events. Iskenderun (Alexandretta) port city is located along the eastern shore of the Gulf of Iskenderun with approximately 152 km of coastline. The modern urban area of Iskenderun is built on marshlands and a coastal alluvial plain on the flanks of the Amanos Mountains. The salt marshes that surrounded the historical city center were drained during the early 20th century. Most of the liquefaction manifestations were detected through a former inundated area that was filled up during the 1980s expansion of the coastal front [49,79] (Figure 11). In particular, Iskenderun City was severely damaged due to widespread lateral spreading and liquefaction along its coastal front fill to the west and central part, while large areas and sections of piers in the port facilities to the east were submerged due to lateral spreading phenomena (Figure 12). Subsidence near the coastal front resulted in shallow flooding of the northernmost part of the city for a few days after the earthquakes [49]. At the same time, Ataturk Boulevard and the coastal front were covered by extensive liquefaction ejecta, visible as light–dark gray sand/silt deposits in post-earthquake VHR satellite imagery.

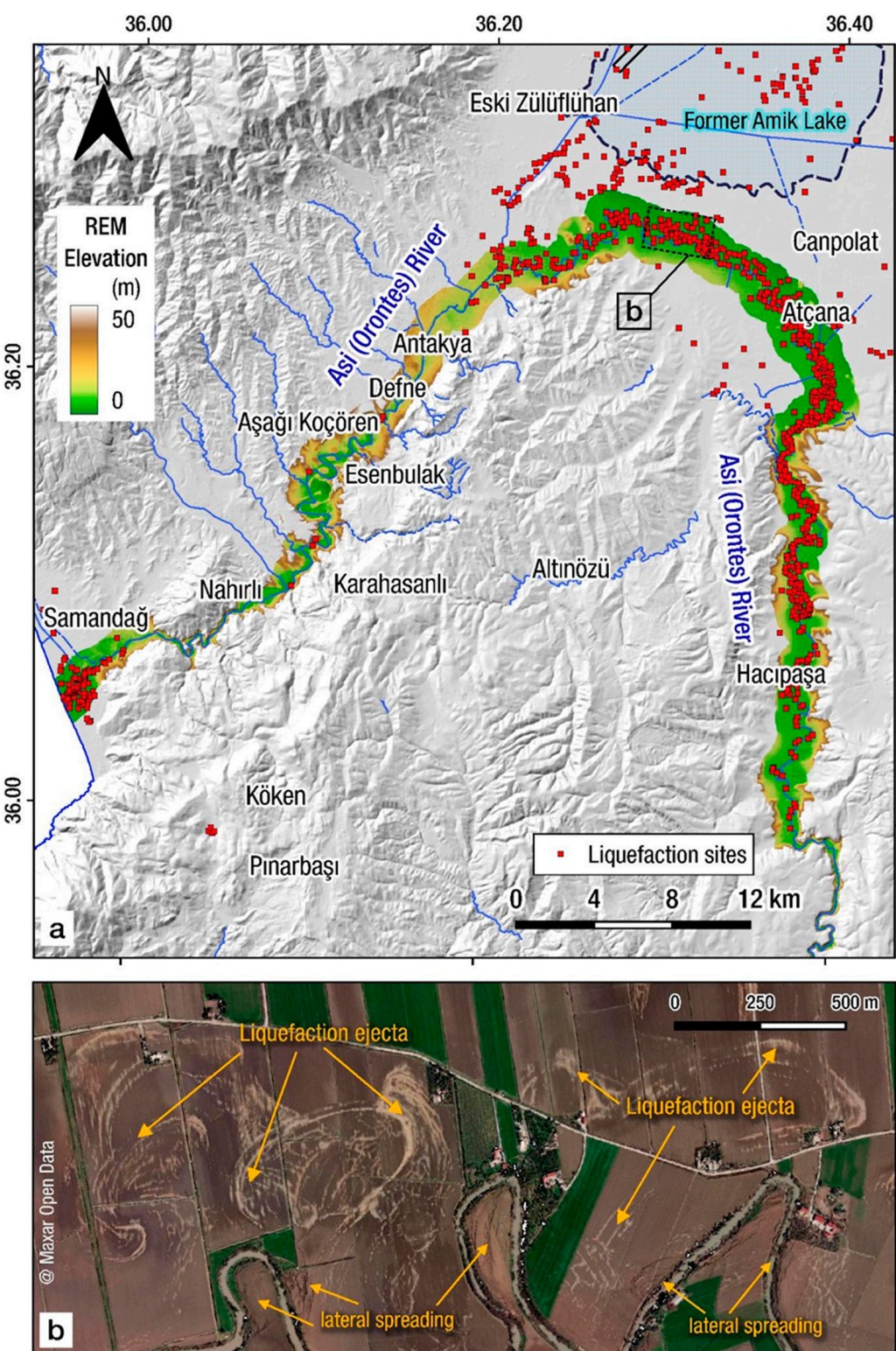

**Figure 10.** (**a**) Relative elevation model (REM) of Orontes River Valley, south of the former Amik Lake. Liquefaction sites are marked with red dots. Elevation source: Copernicus DEM. (**b**) Detection of liquefaction ejecta material (light colors) and lateral spreading phenomena along a meandering section of Orontes River in Maxar VHR satellite imagery.

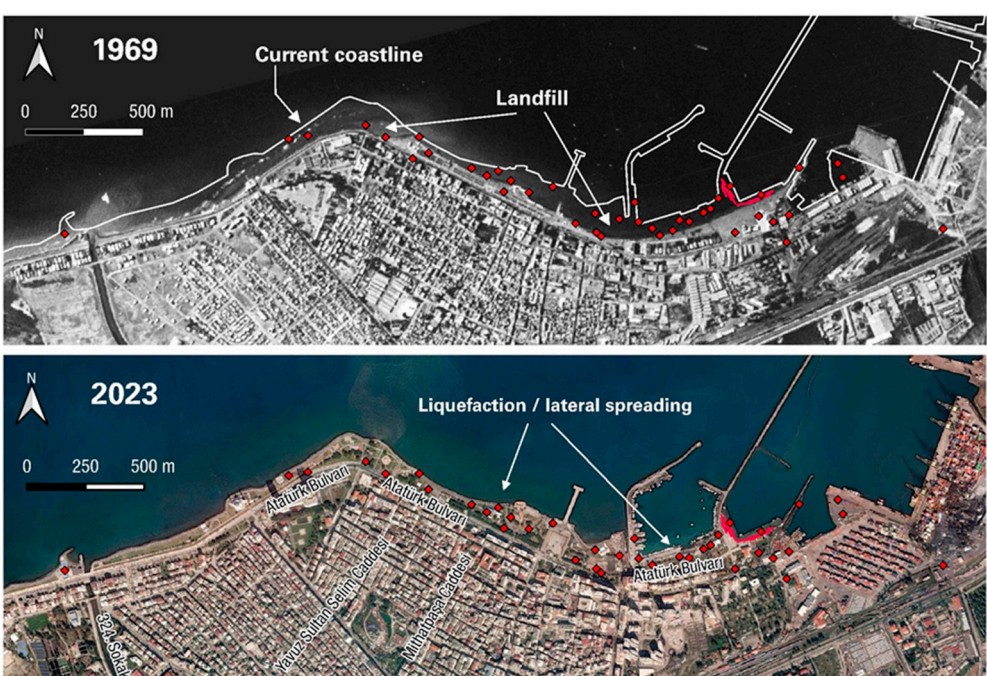

**Figure 11.** (**Above**) KH-4 Corona declassified satellite imagery from 1969, showing the extent of landfill and coastal front expansion during the past decades (current shoreline with white line), (**Bottom**): Post-earthquake VHR optical satellite imagery by Maxar showing the current status in Iskenderun. Liquefaction-related phenomena are marked with red.

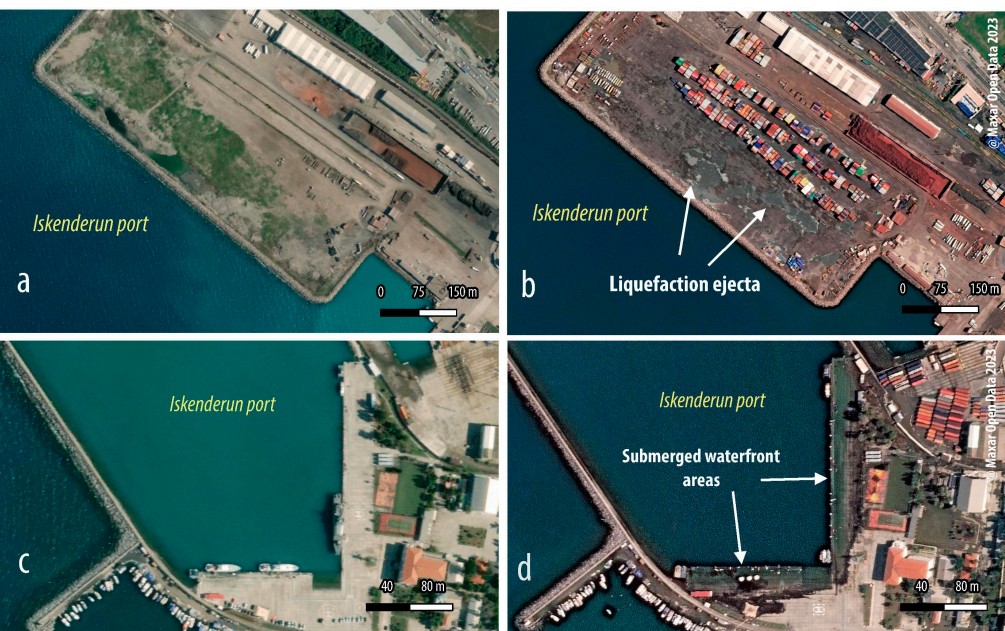

**Figure 12.** Pre-earthquake (**a**,**c**) and post-earthquake VHR satellite imagery (Maxar) showing liquefaction ejecta on the piers of Iskenderun port (**b**) and submersion of specific sections due to lateral spreading (**d**).

Towards the north, extended liquefaction features were observed at the area of Gölbasi Lake, which is part of the pull-apart structure of Gölbasi Basin along with the lakes Inekli and Azapli [80–82]. The main geological units through this region are Late Cretaceous to Plio-Pleistocene age sedimentary rocks. A well-developed drainage system coming from the surrounding high topographic escarpments forms around the basin large alluvial fans,

while the areas between lakes are covered by marshland and irrigated crops. Following the Pazarcik segment of EAF, 173 liquefaction sites were mapped through the Holocene and Quaternary formations of the Gölbasi Basin (Figure 13). Focusing on the Gölbasi Lake and the homonym town, a significant number of extensive lateral spreading phenomena (multiple parallel large fractures) were detected along its northern, eastern, and southern coastline and through the city. It should be pointed out that in some places, large sections of land were totally submerged along the coastline (Figure 14).

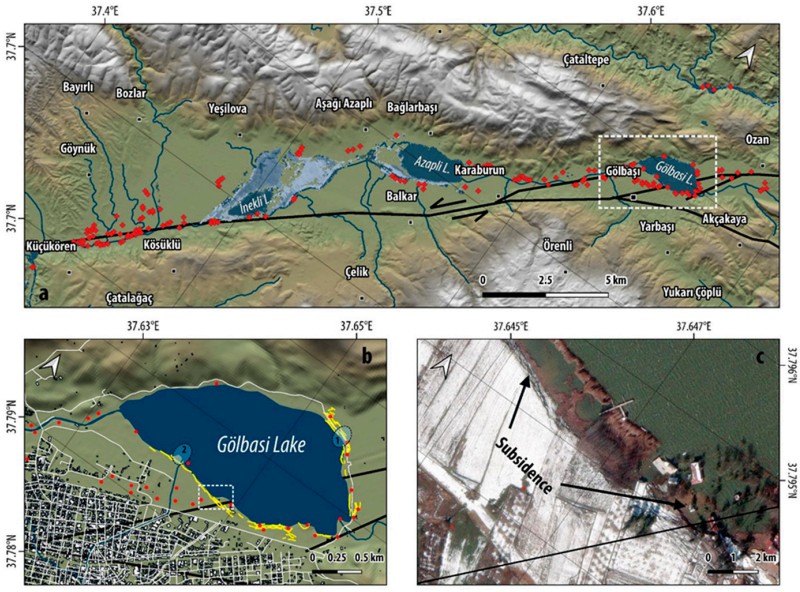

**Figure 13.** (**a**) Distribution of liquefaction phenomena along Gölbasi, Azapli, and Inekli Lakes in the Gölbasi Basin (red markers) with the majority found along the 6 February fault rupture and around the lakes/marshes. (**b**) Lateral spreading deformation and cracks (yellow lines) along the coast of Gölbasi Lake and liquefaction phenomena in the urban area of Gölbasi City. (**c**) Post-earthquake VHR optical satellite imagery by Maxar depicting in detail a submerged section of Gölbasi Lake coast due to lateral spreading.

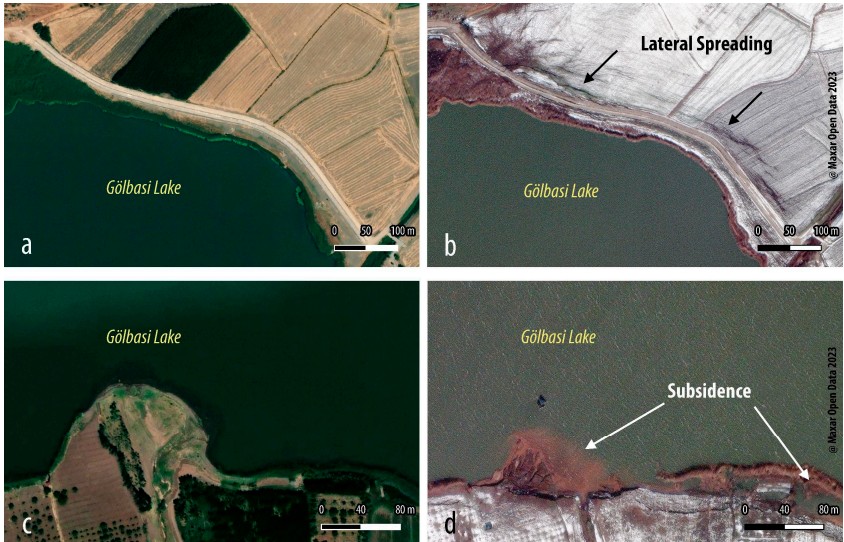

**Figure 14.** Pre-earthquake satellite imagery (**a**,**c**) and post-earthquake satellite imagery (Maxar) showing the lateral spreading phenomena (**b**) and the subsidence of specific areas in the coastline zone of Gölbasi Lake (**d**).

### 3.2. Correlating the Distance of Liquefaction Sites with Location of Epicenters and the Fault Rupture

Taking into account the liquefaction inventory compiled based on information provided by satellite and aerial imagery, we investigated the distribution of liquefaction sites according to their distance from both the fault rupture and the earthquake epicenters. We used the Mw 7.7 epicenter from AFAD, and the surface trace of the fault rupture from Reitman et al. [52].

Considering previous studies [83–86], the expected maximum epicentral distance of liquefied sites was approximately 150 km for an earthquake magnitude M7.7 (Figure 15). According to our liquefaction inventory, this statement is validated since 92% of the total records are distributed within a region 150 km in distance from the epicenter. Regarding the distribution of liquefaction sites in relation to the surface trace of the fault rupture, the majority of the occurrences (95%) were detected within a section of 25 km, with very few sites between 25 and 50 km distance (Figure 16).

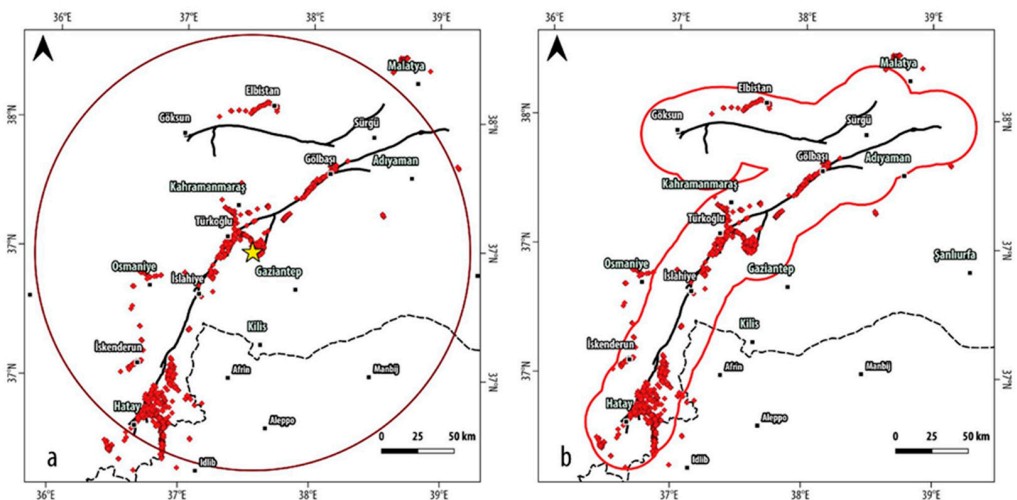

**Figure 15.** Distribution of liquefaction sites in comparison (**a**) with their distance from the epicenter of the 6 February 2023 M7.7 earthquake (yellow star) and (**b**) the fault rupture (black line).

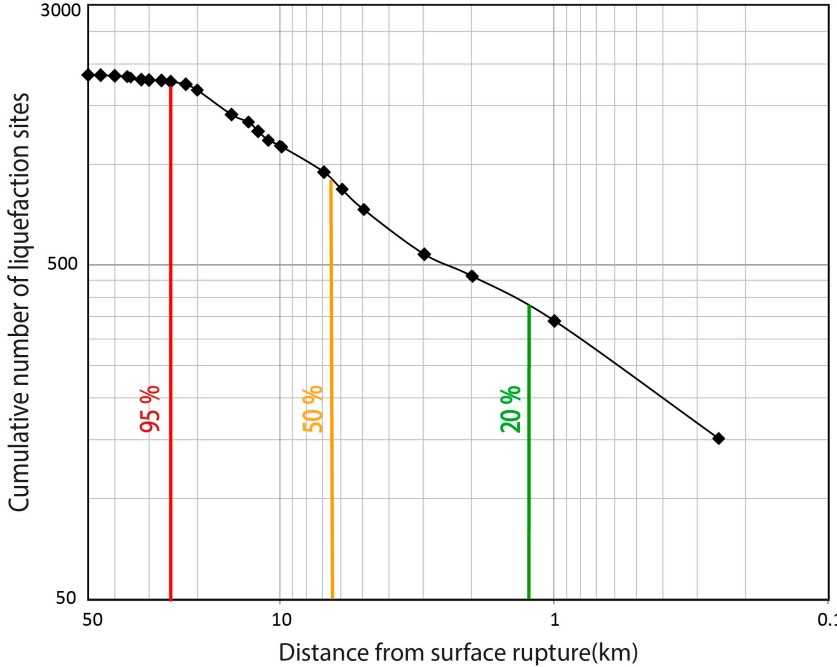

**Figure 16.** Cumulative frequency diagram of distance from the fault rupture for the liquefaction sites.

Consequently, both cumulative distance graph curves follow a declining trend, as the distances from the epicenter and fault rupture increase. However, in the case of the epicenter's curve, dense liquefaction effects are detected through a much broader area (150 km) than those based on the rupture's distance (25 km), implying that distance from the fault rupture of the seismic event is (a) more geographically specified and consequently far less conservative than the epicenter distance and (b) can define more effectively the limits of liquefaction manifestation distribution.

## 4. Discussion

### 4.1. Correlating the Spatial Distribution of Liquefaction Phenomena with the Seismic Parameters and the Geomorphology

The generation of liquefaction phenomena depends on the susceptibility of the sediments and on seismic parameters like earthquake magnitude and strong ground motion, which is strongly related to the distance from the epicenter or fault rupture and the local site conditions [87,88]. As described in the previous section, the majority of liquefaction sites triggered by the 6 February 2023 earthquakes in Turkey/Syria were detected within a certain distance from the fault rupture (25 km). Distance from the fault rupture is a more suitable metric than epicentral distance for assessing liquefaction occurrence in a relevant hazard study, whenever a detailed seismogenic source can be defined, also discussed in compiled magnitude/distance empirical relations [89].

For the purposes of this study, we took into account the ShakeMap [90] of the 6 February 2023 earthquakes, produced by the United States Geological Survey (USGS) to investigate the spatial distribution of liquefaction sites in relation to the peak ground acceleration (PGA) and peak ground velocity (PGV). In particular, an updated version of the original ShakeMap was used (accessed on 26 April 2023) that incorporated a more detailed fault rupture model [91]. The very first versions of the ShakeMap did not include a finite fault rupture model and were considered to be inaccurate as only a simple circular distance from the preliminary epicenter is being incorporated for ground motion attenuation. This is a known issue with ShakeMap products, which cannot incorporate a fault source without human input during the first hours or days after an event. A combined raster map of the maximum PGA and PGV values for both events was produced by stacking of the original products (Figure 17). In this way, we were able to reduce the uncertainties that arose due to possible inaccuracies in fault rupture modeling and a lack of detail for local site conditions, etc., in the attribution of liquefaction occurrences to the two main events.

Distribution analysis and frequency graphs (Figure 17) reveal similar results to distance from the fault rupture; the majority of mapped liquefaction sites are within a narrow zone close to the earthquake ruptures where high values of PGA and PGV are observed. The 95th percentile of liquefaction sites correlates with values of PGA > 0.14 g and PGV > 12 cm/s. These observations are valuable for the early prediction of earthquake environmental effects in future events and for the production of liquefaction hazard maps.

The correlation of liquefaction sites with distance from the fault rupture and the values of strong ground motion (PGA, PGV) defines a geographical boundary to liquefaction occurrence. However, their distribution and concentration within this specific zone are uneven. It is pointed out that not all the low relief areas at a short distance from the fault rupture, covered by recent sediments can manifest liquefaction phenomena; the majority of mapped liquefaction sites formed clusters in specific types of alluvial sediments. The variability of geomorphological settings and surficial sediments within these areas is highly correlated with the severity and clustering of liquefaction. That was the key factor that motivated us to implement the proposed workflow for remote liquefaction mapping.

In general, geomorphology plays a contributory role regarding the spatial distribution of liquefaction phenomena [4,12,22,24,25,29,92]. As described in previous sections, the majority of liquefaction effects were concentrated along the river network in meander loops (inner part of the meander), abandoned or recently in-filled river strands as well as in drained lake basins and coastal zones.

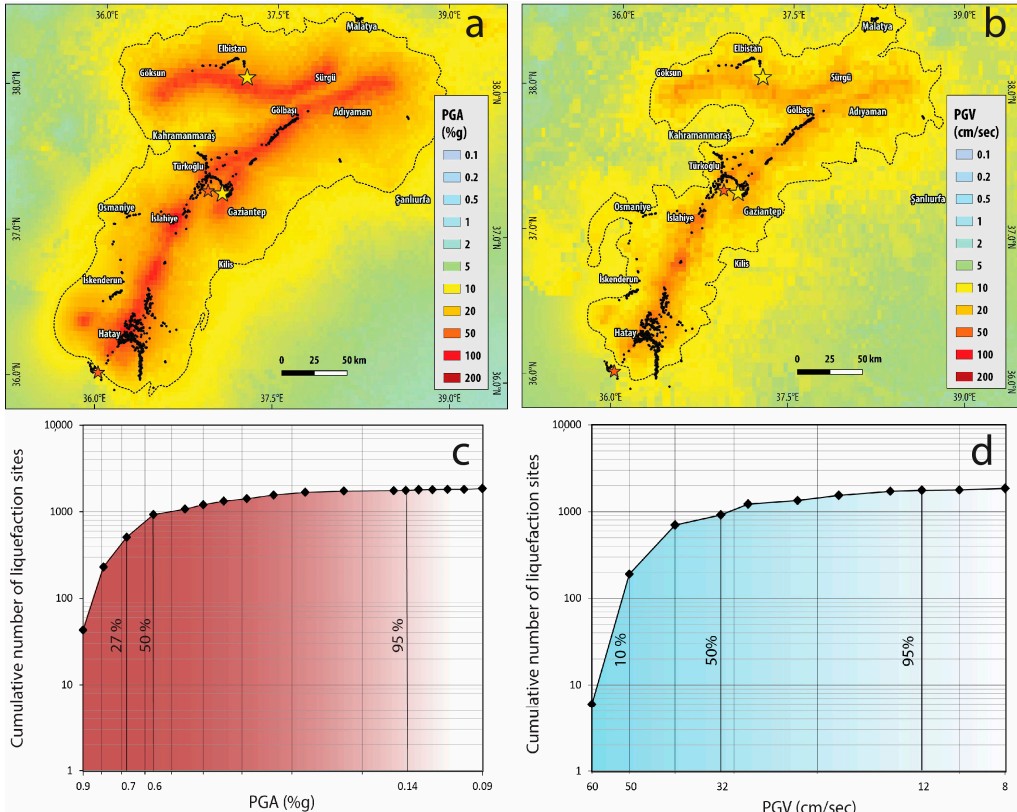

**Figure 17.** Distribution of liquefaction sites in comparison (**a**) with peak ground acceleration ShakeMap and (**b**) peak ground velocity ShakeMap. Cumulative frequency diagrams for PGA (**c**) and PGV (**d**). Contour of 0.14 g PGA and contour of 12 cm/s PGV that represent the 95th percentile of liquefaction sites are shown in (**a**,**b**) as a dotted line. Yellow stars show the epicenters of the two mainshocks (M7.7 and M7.6) and orange stars the location of major aftershocks (M6.7 and M6.4).

One of the areas where a large number of liquefaction and lateral spreading occurrences were mapped, is near the central section of the Mw 7.7 rupture and its epicenter. Aksu River is forming a bend west of Pazarcik between the Pazarcik segment of EAF to the west and the Narli Fault zone to the south, crossing through Saglik and Narli Basins (Figure 18). These basins are filled with Quaternary alluvial sediments, covered by the Holocene river, floodplain, and lake deposits [54]. At several locations, the Aksu River course has been shifted, especially in Narli Basin, as can be observed through morphology and declassified historical satellite imagery (Corona KH-4). Most of the mapped liquefaction sites in this area were found along abandoned stretches of Aksu River, migrated meander sections (see Figure 18 insets), drained swamps, and small lakes inside Narli Plain and along the EAF and Narli Fault earthquake surface ruptures. Lateral spreading was observed in numerous drainage and irrigation canals within Narli Plain. In the middle and southern part of Narli Basin, numerous ground fractures unconnected to free face slopes and with random orientation were mapped. We interpret this as evidence of extensive ground oscillation deformation, possibly related to drained lakes and swamps. Figure 18 shows the distribution of liquefaction sites over the USGS near a real-time liquefaction hazard map [93]. Almost all the sites are within the high probability area correctly predicted by the USGS model. However, the real-time map overpredicts probable liquefaction occurrence due to a lack of detailed data about surficial geology and geomorphology.

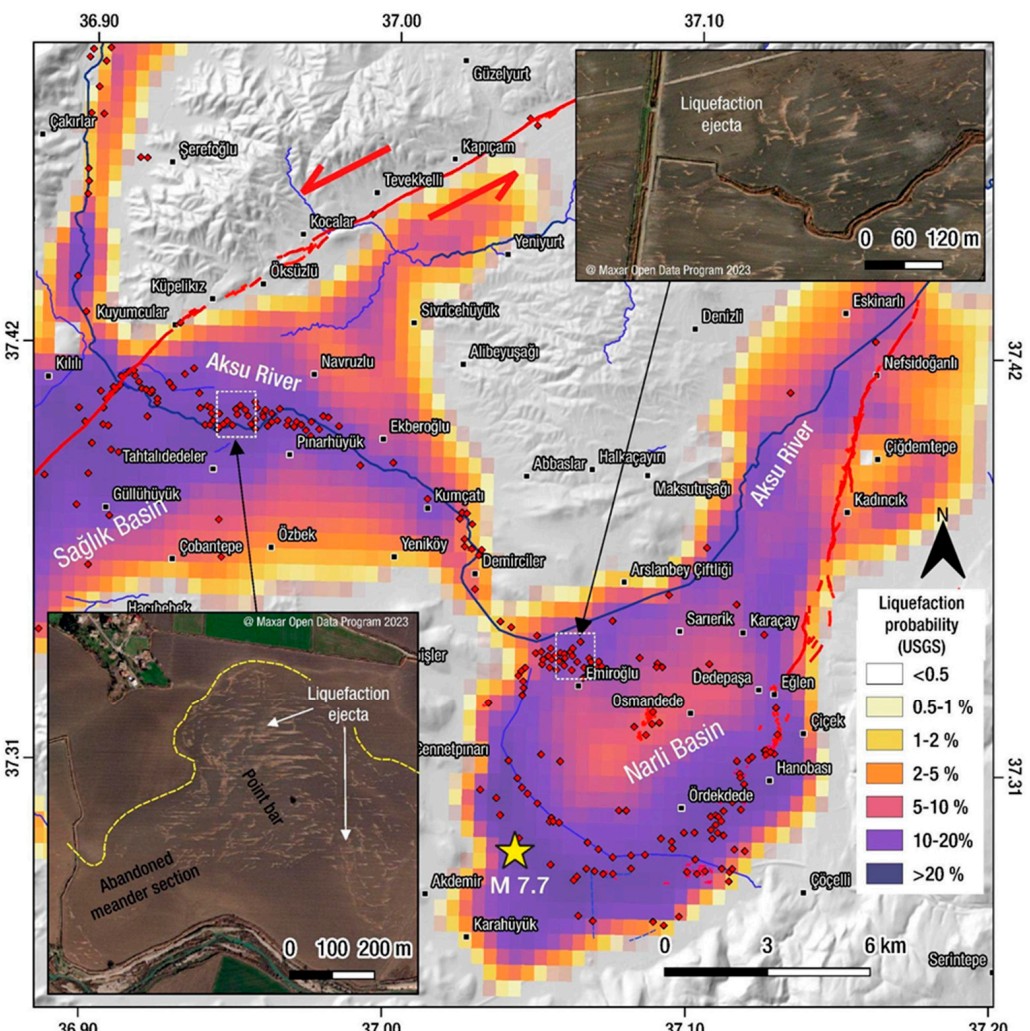

**Figure 18.** Liquefaction and lateral spreading sites (red dots) in Aksu River bend valleys area, near the epicenter of the Mw 7.7 earthquake (yellow star). Ground cracks and earthquake surface ruptures shown as red lines. Insets show details of liquefaction ejecta over paleo-meanders of Aksu River (Maxar Open Data VHR satellite imagery). USGS near real-time liquefaction hazard map as a reference layer.

The most prominent area of liquefaction manifestations during the 6 February 2023 earthquakes was the Hatay region. Numerous liquefaction and lateral spreading sites are detected in the area of Antakya, Orontes River Valley, and Amik (Amuq) Plain (see description in Section 4.1 and Figure 19). Amik Plain is fed by the Karasu River from the north, the Afrin River from the east, and drained by Orontes (Asi) River to the Mediterranean through Antakya Valley. Orontes River is flowing into Amik Plain from the south, following Turkey–Syria border, and exits through the Antakya Valley corridor, creating a large-scale bend (Figure 10). At the eastern and central section, the Orontes (Asi) River forms a wide floodplain valley with modern and abandoned meandering sections. To the west, south of Antakya City, runs through an entrenched course and exits into the Mediterranean. A large part of central western Amik Plain was once covered by Amik Lake (Lake of Antioch), which was completely drained during the 1970s [77,78]. The shoreline of the lake during the 20th century is shown in Figure 19, but the late historic extent of the lake is believed to be larger (Figure 19). Thus, we adopted the contour line of 79–80 m as a more representative paleo-shoreline.

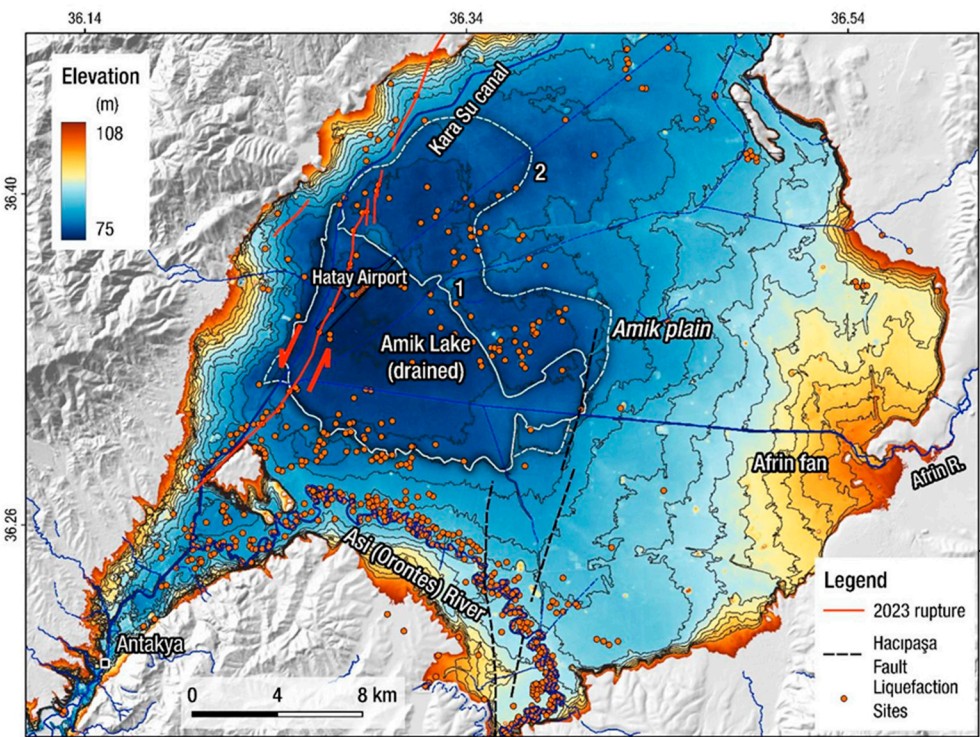

**Figure 19.** Topography of Amik (Amuq) Plain. White continuous line (1) marks the 19th–20th century extent of the now-drained Amik Lake, while white dashed line (2) shows the estimated furthest extent of the historical lake coastline. The 6 February 2023 liquefaction and lateral spreading sites shown as orange dots, and earthquake fault rupture with red lines. Contour line interval is 2 m. Elevation source: Copernicus DEM.

As described in the previous chapter, multiple sand craters, fissures with ejecta material, and lateral spreading phenomena were mapped along meanders, paleochannels, and point-bars formations of the Orontes River Valley (Figure 19). Liquefaction and lateral spreading manifestation and deformation were almost continuous along the Orontes River meandering course from the Turkey–Syria border up to the NE of Antakya City. These phenomena were limited through Antakya Valley due to the entrenched course of Orontes. A large number of liquefaction sites were also mapped along the Kara Su River exit into Amik Plain. The rest of the mapped liquefaction and lateral spreading sites in the area were spread over the drained Amik Lake floor (Figures 19 and 20a) and are related to lateral spreading along drain and irrigation canals, road and flood embankments, and small fluvial features.

A large number of ground cracks with significant length were observed over the former Amik Lake floor. To the west, surface fault ruptures related to a previously poorly known segment of EAF cross Amik Plain through the Hatay airport. However, ground cracks and fractures further east of the rupture are related to liquefaction/shaking deformation (ground oscillation–lateral spreading) and/or possibly triggered shallow slip of pre-existing neotectonic fault traces. Some of these ground cracks were interpreted as primary or secondary fault ruptures [52]. Nevertheless, a large number of those display random orientations, more similar to lateral spreading and ground oscillation deformation, while others follow the concave trace of multiple paleo-shorelines, visible in terrain and historical imagery to the north. Additionally, all ground cracks that were mapped east of the EAF rupture and Hatay airport are completely contained in the paleo-Amik Lake extent (Figure 19) with no evidence of continuation to the surrounding alluvial and fan sediments. This could be a strong indication of ground oscillation across former Amik Lake, expressed as deformation of the lake floor sediment cover.

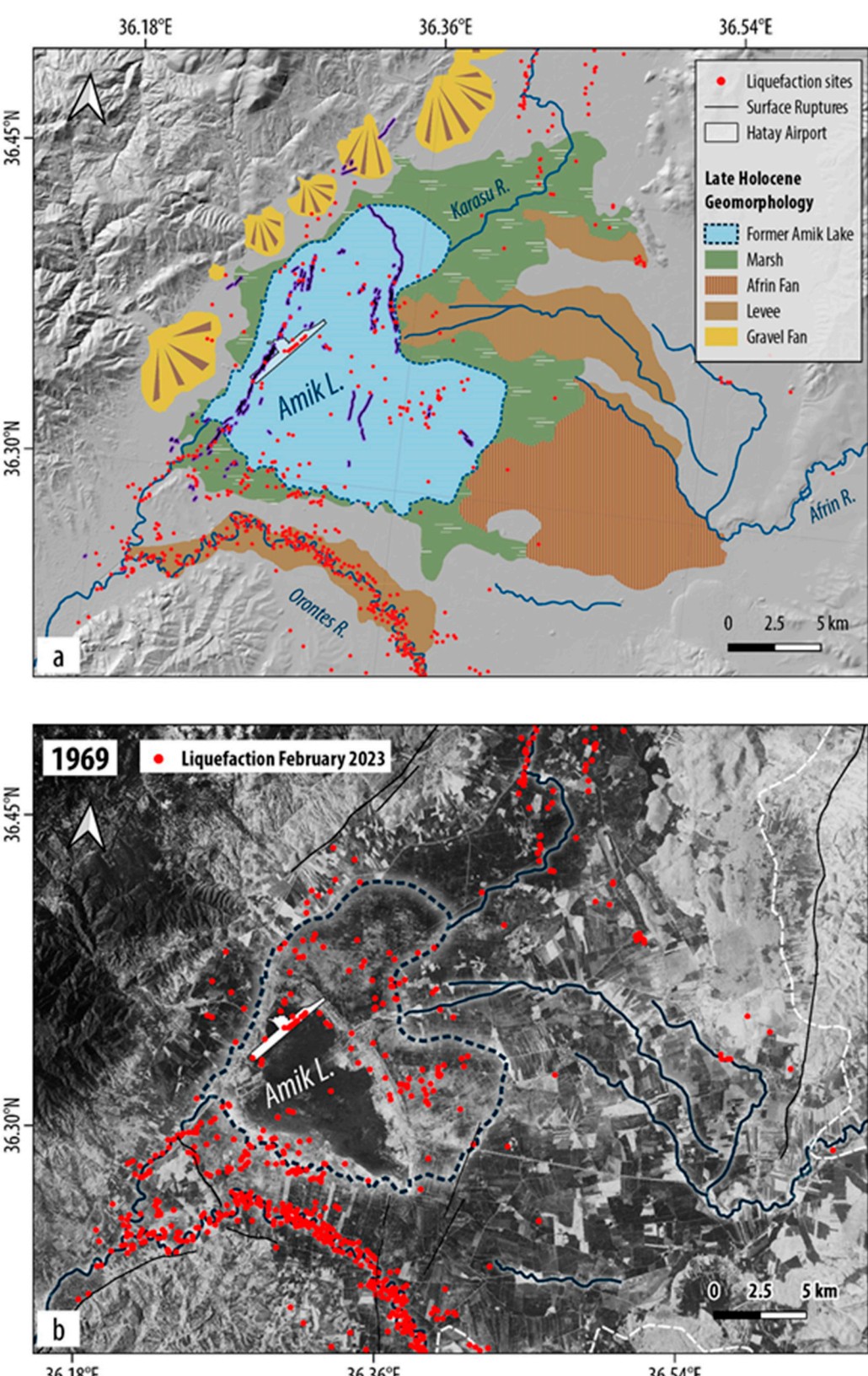

**Figure 20.** (**a**) Late Holocene geomorphology of Amik (Amuq) Plain from [74], with liquefaction and lateral spreading sites (red dots) and surface ruptures/ground cracks (purple lines). (**b**) Liquefaction and lateral spreading sites (red dots) over Amik (Amuq) Plain over a declassified KH-4 Corona satellite imagery from 1969. Black dotted line marks the estimated furthest extent of historical Amik Lake.

The overall distribution of liquefaction and lateral spreading sites in Amik Plain is strongly correlated to the late Holocene geomorphological conditions as can be seen in Figure 20a. Geomorphological units of Amik Plain [72] that manifested severe liquefaction phenomena were the former Amik Lake and the fluvial valley/levee of Orontes River. Despite modern-era modification of ground surface conditions by large-scale irrigation and drainage works, Holocene and historical geomorphological features can be identified through historical imagery and detailed digital terrain models (Figures 19 and 20b).

### 4.2. Recurrent Liquefaction in Antakya

A few days after the main shock (20 February), a strong earthquake of Mw 6.4 occurred near Antakya in the Hatay region. This earthquake originated on a NE–SW left-lateral fault plane, dipping towards NW that roughly coincides with the local Antakya Fault [43]. This strong aftershock led to heavy damage in the Antakya urban area, especially in buildings already degraded by the main shock [49]. Sentinel-2 images acquired after the earthquake (24 February) did not reveal any significant change or visual indications of liquefaction at the alluvial plain near Antakya City and Orontes River Valley. We processed two couples of Sentinel-1 frames (ascending and descending orbit) that cover the aftershock timeframe (Figure 21). The interferogram shows a large-scale deformation pattern of concentering fringes that are attributed to the crustal deformation caused by the Mw 6.4 rupture. The reversed polarity of the fault deformation fringes in the ascending and descending interferograms matches with mostly horizontal deformation, consistent with the fault mechanism by AFAD. Along the Antakya Valley, a series of distinct and irregular fringe patterns and low coherence patches are visible in the phase interferograms (Figure 21). These small patterns are persistent in both interferometric pairs, excluding atmospheric or other types of error, and of very shallow origin overlapping the large-scale fault rupture deformation. We attribute them to (a) deep-seated landslide displacement triggered by the aftershock in the hilly area northwest of Antakya, and (b) liquefaction and lateral spreading deformation along the Orontes alluvial plain to the northeast of Antakya. The same area just outside Antakya urban area also experienced multiple liquefaction manifestations during the mainshock on 6 February.

Recurrent liquefaction and lateral spreading occurrence were also identified through satellite interferometry in the 2010–2011 Christchurch, NZ earthquake sequence [16] and the 2021 Thessaly, Greece earthquakes [12]. This highlights the importance of remote sensing methods and mapping for the rapid identification of liquefaction and other types of earthquake environmental effects. The occurrence of strong earthquakes a few days or weeks after a significant event can create an amalgamation of liquefaction and lateral spreading phenomena from multiple events that can only be unrolled through satellite imagery and aerial acquisitions.

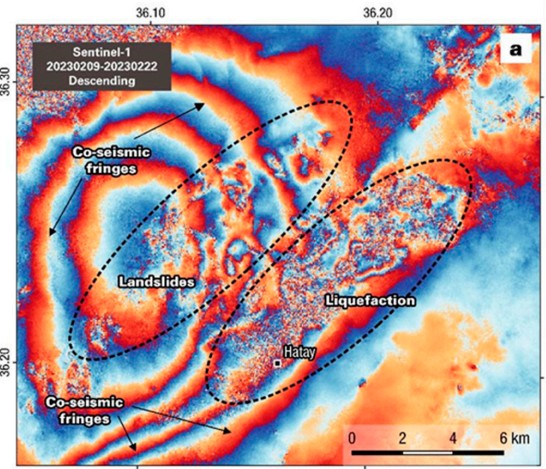

**Figure 21.** *Cont.*

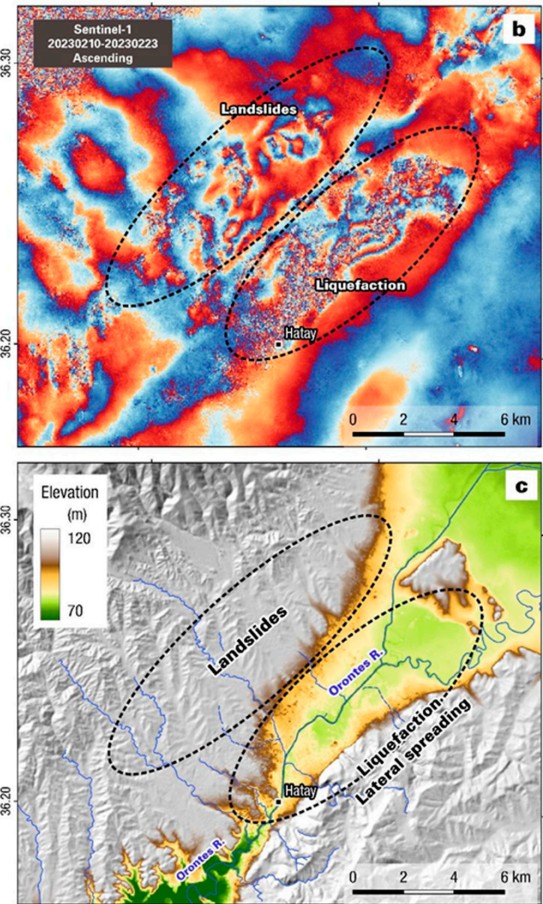

**Figure 21.** Recurrent liquefaction from the 20 February Mw 6.4 strong aftershock in Antakya area. Irregular features (isolated fringe patterns and spotted areas with loss of coherence) that are persistent in both ascending and descending interferograms (**a,b**) and not related to the larger scale fault deformation patterns (larger fringes in (**a,b**)) correspond to areas with widespread landslides and slow-moving landslide deformations (northern dashed ellipse) across the hills opposite of Antakya City, and widespread liquefaction and lateral spreading deformation (southern dashed ellipse) across the meandering valley of Orontes River just north of Antakya (**c**).

## 5. Conclusions

We used satellite imagery to map 1850 sites with liquefaction and lateral spreading phenomena triggered by the 6 February 2023 Mw 7.7 and Mw 7.6 earthquakes in Turkey/Syria. We applied and further improved a methodology used by [12] for rapid mapping of earthquake-induced liquefaction.

(1) High and very high-resolution optical satellite imagery, with the aid of radar satellite imagery and interferometry, enabled us to rapidly acquire a thorough map of liquefaction manifestations and sites across a large area affected by the two strong earthquakes.

(2) Despite the limitations of the current mapping data and results, we consider this map an almost complete documentation of liquefaction site distribution across the affected area in SE Turkey and Syria. Most areas with significant concentration and severe magnitude of liquefaction phenomena are also described.

(3) Application of the proposed workflow for mapping liquefaction with remote sensing data successfully limited the search focus for this study and enabled rapid mapping through a vast area.

(4) The majority of liquefaction phenomena were found along meandering sections of river valleys, coastal plains with fine sediments, drained lakes and swamps, and lacustrine basins along the East Anatolian Fault. In addition, it is crucial to highlight

the very high liquefaction susceptibility of reclaimed lands as it has been confirmed in the case of the city of Iskenderum and Hatay airport. These areas are suggested to be studied in detail in order to decrease the liquefaction potential of the subsoil layers and consequently minimize the relevant risk to the manmade environment.

(5) Results confirm once again the major correlation between geomorphology/surficial geology and liquefaction manifestation of strong earthquakes. These geomorphological conditions can serve as a successful proxy of local site investigations at a regional scale and can drive future investigations and focus areas of interest for future liquefaction hazard mapping.

**Supplementary Materials:** The following supporting information can be downloaded at: https://www.mdpi.com/article/10.3390/rs15174190/s1, Figure S1: Flooded areas in Hatay Province after the 6 February 2023 earthquakes, Figure S2: Coverage map of Planetscope optical imagery (Planet) used for earthquake-triggered liquefaction mapping, Figure S3: Coverage map of very high-resolution (VHR) optical imagery (Maxar Open Data) used for earthquake-triggered liquefaction mapping, Figure S4: Coverage map of Sentinel-1 radar satellite imagery frames (Copernicus) used for earthquake-triggered liquefaction mapping, Table S1: Frame id and dates of Copernicus Sentinel-1 SLC frames, Table S2: Frame id and dates of Corona (KH-4B) declassified satellite imagery. Mapped liquefaction and lateral spreading sites are provided in a Geopackage file (Liquefaction_Lateral_Sites_20230206EQ.gpkg).

**Author Contributions:** Conceptualization, S.V. and G.P.; investigation, S.V., M.T. and E.K.; methodology, M.T. and S.V.; data curation, S.V. and E.K.; writing—original draft preparation, M.T. and S.V.; writing—review and editing, M.T., S.V. and G.P.; visualization, S.V. and M.T.; supervision, G.P. and S.V. All authors have read and agreed to the published version of the manuscript.

**Funding:** This research received no external funding.

**Data Availability Statement:** Planetscope imagery was provided by Planet Team (2023); Planet Application Program Interface, In Space for Life on Earth. San Francisco, CA. https://api.planet.com/Planet (accessed on 5 May 2023) (Planet Labs, Inc., San Francisco, CA, USA). Copernicus Sentinel-1 and Sentinel-2 imagery was accessed through Copernicus Open Access Hub (https://scihub.copernicus.eu/ (accessed on 5 May 2023)). Copernicus Global Digital Elevation Model, distributed by OpenTopography (https://doi.org/10.5069/G9028PQB (accessed on 5 May 2023)). WorldView and GeoEye-1 imagery was accessed through Maxar's Open Data Program (https://www.maxar.com/open-data/turkey-earthquake-2023 (accessed on 5 May 2023)).

**Conflicts of Interest:** The authors declare no conflict of interest.

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
