# Peer review of "Satellite Imagery for Rapid Detection of Liquefaction Surface Manifestations: The Case Study of Türkiye–Syria 2023 Earthquakes"

_remotesensing, doi:10.3390/rs15174190_

Round 1

Reviewer 1 Report

Taftsoglou et al. presented a very interesting paper about the detection and investigation of soil liquefaction caused by the doublet (catastrophic) recent earthquakes that occurred on 6 February 2023 in Turkey close to the border with Syria. The paper used remote sensing satellite data, such as Sentinel 2 visual maps and higher resolution satellite products as well as radar satellite data from Sentinel 1. The satellite methodology used by the author is reliable and well-used in the field including the application of official satellite data processing toolbox such as SNAP by ESA. The authors carefully investigate the data to avoid automatic problems related to the snow that was presented at the time of the earthquake in larger high-elevation areas in the region. They present the results in a deeply discussed geology and geomorphology of the ground. The results support the conclusion, which is that the liquefaction sites are better related to distance from the surface rupture rather than the epicenter; the liquefactions strongly occurred in correspondence with drained or ancient lakes and, in addition, along the rivers of the region such as Orontes along the border with Syria. In addition, the authors elaborated InSAR from Sentinel 1 data and discussed further liquefaction caused by a large aftershock on 20 February 2023 of magnitude Mw6.4. Furthermore, the paper is very well written. For all of these reasons, I recommend this paper to be published in Remote Sensing after a few minor points, which I list below.

Finally, reading your paper, I had the impression that larger liquefaction of ground with, consequently, damages to human structures occurred in correspondence with artificially modified ground (drained) to build new constructions. This poses a further warning, in my opinion, for future constructions to be more in agreement with nature and not force the natural landscape, avoiding constructions in high-risk hydrogeological and seismic hazard places. If the authors agree with their discretion, they can spend some words in the final discussion or conclusion on this point.

Specific/minor points:

·        Line 13. I think it’s better to specify “soil liquefaction”

·        Line 49. I think “sporadically due” is better

·        Line 70. I suggest adding the first name of the refence even if not required by journal stile to be more readable: Ramakrishnan et al. [18] used... the same at line 73, 85

·        Lines 102-103. Is there a gross estimation of the magnitude of these historical earthquakes? Even though it would be associated with a large uncertainty, it’s better to include an estimation if exist.

·        Line 133. Maybe it’s better say “different fault structures” if you mean this.

·        Line 138. To avoid the second “buildings”, maybe it can be substituted with “ones”. In addition, in the end, I think it’s better to use past tense (reached, not reaches).

·        Line 221. I think it’s more proper to say “possible liquefaction”

·        Line 237. Please, in this line, I would like you to say exactly the time you analysed not “few weeks after the event”.

·        Line 255. I would suggest to underline that Planet is a service or using italic and quotation marks or write explicitly “Planet service”

·        Line 278. Please add the time interval that you analysed with Sentinel 1A InSAR

·        Line 311. Please put superscript the 2 (km2). The same at line 484.

·        Figure 16. The graph seems wrong with respect to the description. In fact the first point would means that within 0.1 km there are about 2000 liquefaction sites and going far they decreased. If this is the binned number of liquefaction sites this is right, but the lable of vertical axis is wrong. If the label is right so it’s the “cumulative” number then going far the curves must increase not decrease. Please check and revise according. The same applies to Figure 17c and d.

·        Line 603. Maybe it’s better to remark that is a “more suitable metric” than epicentral distance.

·        Discussion about shake maps. Even though I totally agree and support the choice of the authors to use updated Shake Maps for this paper, it would be interesting to have an idea of what could be the impact of using early seismological products, thinking to a future application of your method in quasi-real time so before refined products are available. If the authors can add a short discussion/sentence, it would be appreciable.

Author Response

Taftsoglou et al. presented a very interesting paper about the detection and investigation of soil liquefaction caused by the doublet (catastrophic) recent earthquakes that occurred on 6 February 2023 in Turkey close to the border with Syria. The paper used remote sensing satellite data, such as Sentinel 2 visual maps and higher resolution satellite products as well as radar satellite data from Sentinel 1. The satellite methodology used by the author is reliable and well-used in the field including the application of official satellite data processing toolbox such as SNAP by ESA. The authors carefully investigate the data to avoid automatic problems related to the snow that was presented at the time of the earthquake in larger high-elevation areas in the region. They present the results in a deeply discussed geology and geomorphology of the ground. The results support the conclusion, which is that the liquefaction sites are better related to distance from the surface rupture rather than the epicenter; the liquefactions strongly occurred in correspondence with drained or ancient lakes and, in addition, along the rivers of the region such as Orontes along the border with Syria. In addition, the authors elaborated InSAR from Sentinel 1 data and discussed further liquefaction caused by a large aftershock on 20 February 2023 of magnitude Mw6.4. Furthermore, the paper is very well written. For all of these reasons, I recommend this paper to be published in Remote Sensing after a few minor points, which I list below.

Finally, reading your paper, I had the impression that larger liquefaction of ground with, consequently, damages to human structures occurred in correspondence with artificially modified ground (drained) to build new constructions. This poses a further warning, in my opinion, for future constructions to be more in agreement with nature and not force the natural landscape, avoiding constructions in high-risk hydrogeological and seismic hazard places. If the authors agree with their discretion, they can spend some words in the final discussion or conclusion on this point.

We would like to thank reviewer for his/her comments. A comment for manmade fill areas with liquefaction occurrence was added in the conclusions section.

Line 13. I think it’s better to specify “soil liquefaction”

We rephrased the term in the abstract.

Line 49. I think “sporadically due” is better.

We corrected phrasing in the text.

Line 70. I suggest adding the first name of the reference even if not required by journal stile to be more readable: Ramakrishnan et al. [18] used... the same at line 73, 85

We rephrased the relevant references in the text.

Lines 102-103. Is there a gross estimation of the magnitude of these historical earthquakes? Even though it would be associated with a large uncertainty, it’s better to include an estimation if exist.

We added magnitudes for the historical earthquakes in the text.

Line 133. Maybe it’s better say “different fault structures” if you mean this.

We corrected phrasing in the text.

Line 138. To avoid the second “buildings”, maybe it can be substituted with “ones”. In addition, in the end, I think it’s better to use past tense (reached, not reaches).

We corrected phrasing in the text.

Line 221. I think it’s more proper to say “possible liquefaction”

We corrected the term in the text.

Line 237. Please, in this line, I would like you to say exactly the time you analysed not “few weeks after the event”.

The exact time frame was added in the text.

Line 255. I would suggest underlining that Planet is a service or using italic and quotation marks or write explicitly “Planet service”

We explicitly describe the imagery source of Planet data (Planetscope constellation) thus we consider that is a more proper way for describing those data sources than the general term Planet service (which might include other sources and type of imagery).

Line 278. Please add the time interval that you analyzed with Sentinel 1A InSAR

Sentinel-1 frames and dates are included in the supplementary section for the reader to estimate the time interval for all pairs analyzed. Multiple time intervals were investigated that include co-seismic interferograms and post-seismic interferograms (including the ones covering the Mw 6.4 aftershock in Antakya). We added though the acquisition interval for Sentinel-1 pairs (12 days) that defines the minimum time frame of each possible InSAR pair.

Line 311. Please put superscript the 2 (km2). The same at line 484.

We corrected the relevant section.

Figure 16. The graph seems wrong with respect to the description. In fact the first point would means that within 0.1 km there are about 2000 liquefaction sites and going far they decreased. If this is the binned number of liquefaction sites this is right, but the label of vertical axis is wrong. If the label is right so it’s the “cumulative” number then going far the curves must increase not decrease. Please check and revise according. The same applies to Figure 17c and d.

We corrected Figures 16 and 17c and d.

Line 603. Maybe it’s better to remark that is a “more suitable metric” than epicentral distance.

We corrected phrasing in the text.

Discussion about shake maps. Even though I totally agree and support the choice of the authors to use updated Shake Maps for this paper, it would be interesting to have an idea of what could be the impact of using early seismological products, thinking to a future application of your method in quasi-real time so before refined products are available. If the authors can add a short discussion/sentence, it would be appreciable.

We added more comments in the text about the issues with early ShakeMap products.

Reviewer 2 Report

I think it's a really good research paper to study the rapid detection of liquefaction surface manifestations of Türkiye-Syria 2023 earthquakes. Based on visual mapping with optical satellite imagery and the aid of radar satellite imagery and interferometry, a thorough map of earthquake triggered liquefaction is acquired. 1850 sites with liquefaction manifestation and lateral spreading deformation are identified in the paper. Just like the author said, I think this map is an almost complete documentation of liquefaction site distribution across the affected area in SE Turkey and Syria. It will contribute to improving emergency response and minimize in duced casualties, also driving future investigations and focus areas of interest for future liquefaction hazard mapping. Some of the conclusions are impressive. I believe this paper will be of interest to a wide range of readers.

Some details need to be revised.

Lines 163-200:A sketch map is needed to show those faults, tectonics, and locations.

The lower right map in Figure 2 has illegible writing.

Line 234: It should be 2.1.1.

Line 274: It should be 2.1.2.

Line 286: It should be 2.1.3.

Line 307: It should be 2.2.

Line308: It should be 2.2.1.

Line 299-300: Frame id and dates of Corona frames used are included in the supplementary section. I didn’t see it in the supplementary section.

Line 346: It should be 2.2.2.

Line 390: It should be 2.2.3.

The following sequence numbers of subsections are wrong.

Line 601: February 20th 2023 earthquakes?

Author Response

I think it's a really good research paper to study the rapid detection of liquefaction surface manifestations of Türkiye-Syria 2023 earthquakes. Based on visual mapping with optical satellite imagery and the aid of radar satellite imagery and interferometry, a thorough map of earthquake triggered liquefaction is acquired. 1850 sites with liquefaction manifestation and lateral spreading deformation are identified in the paper. Just like the author said, I think this map is an almost complete documentation of liquefaction site distribution across the affected area in SE Turkey and Syria. It will contribute to improving emergency response and minimize in induced casualties, also driving future investigations and focus areas of interest for future liquefaction hazard mapping. Some of the conclusions are impressive. I believe this paper will be of interest to a wide range of readers.

We would like to thank reviewer for his/her comments.

Lines 163-200:A sketch map is needed to show those faults, tectonics, and locations.

We show the names of main fault segments of EAFZ and regional faults in Figure 2. Missing one were added in Figure 2 inset map (regional structures).

The lower right map in Figure 2 has illegible writing.

We improved labeling in Figure 2 inset map.

Line 234: It should be 2.1.1.

Line 274: It should be 2.1.2.

Line 286: It should be 2.1.3.

Line 307: It should be 2.2.

Line308: It should be 2.2.1.

We corrected sequence numbering of the text sections and subsections.

Line 299-300: Frame id and dates of Corona frames used are included in the supplementary section. I didn’t see it in the supplementary section

Frame ids and dates of Corona (KH-4B) frames used are now included in the supplementary section (Table S2).

Line 346: It should be 2.2.2.

Line 390: It should be 2.2.3.

We corrected sequence numbering of the text sections and subsections.

The following sequence numbers of subsections are wrong.

Line 601: February 20th 2023 earthquakes?

We corrected the date in the text.
